# Giant electrically tunable magnon transport anisotropy in a van der Waals antiferromagnetic insulator

Shaomian Qi[1,10], Di Chen [2,10], Kangyao Chen[1], Jianqiao Liu[1], Guangyi Chen[1], Bingcheng Luo[1], Hang Cui[1], Linhao Jia[1,2], Jiankun Li[2], Miaoling Huang[2], Yuanjun Song[2], Shiyi Han[3], Lianming Tong [3], Peng Yu[4], Yi Liu[5], Hongyu Wu[6], Shiwei Wu [7], Jiang Xiao [7], Ryuichi Shindou[1], X. C. Xie[1,8] & Jian-Hao Chen [1,2,8,9] ✉

Anisotropy is a manifestation of lowered symmetry in material systems that have profound fundamental and technological implications. For van der Waals magnets, the two-dimensional (2D) nature greatly enhances the effect of in-plane anisotropy. However, electrical manipulation of such anisotropy as well as demonstration of possible applications remains elusive. In particular, in-situ electrical modulation of anisotropy in spin transport, vital for spintronics applications, has yet to be achieved. Here, we realized giant electrically tunable anisotropy in the transport of second harmonic thermal magnons (SHM) in van der Waals anti-ferromagnetic insulator $CrPS_4$ with the application of modest gate current. Theoretical modeling found that 2D anisotropic spin Seebeck effect is the key to the electrical tunability. Making use of such large and tunable anisotropy, we demonstrated multi-bit read-only memories (ROMs) where information is inscribed by the anisotropy of magnon transport in $CrPS_4$. Our result unveils the potential of anisotropic van der Waals magnons for information storage and processing.

Low structural symmetry in materials naturally leads to anisotropy, which is of great interest in fundamental sciences and has abundant applications in daily life. Van der Waals materials, in this regard, are naturally low symmetry systems due to the reduced dimensions. The anisotropic properties in these 2D materials are more prominent and have attracted considerable attentions recently. Anisotropic resistance in non-magnetic 2D materials such as GaTe, $ReS_2$ and black phosphorus has been used to make floating gate memories[1], digital inverters[2] and synaptic devices[3]. Anisotropic light-matter interactions in 2D materials also facilitate applications in various optoelectronic devices like novel polarizers or polarization sensors[4], polarized light-emitting diodes and polarization-sensitive photodetectors[5,6]. In the field of van der Waals magnets, anisotropic excitons in $NiPS_3$ [7], anisotropic optical response in $CrOCl$[8], as well as anisotropic resistance in $CrSBr$[9] and $FeOCl$[10] have been reported, while a demonstration of possible applications is still lacking. So far, highly tunable 2D magnons

[1]International Center of Quantum Materials, School of Physics, Peking University, Beijing, China. [2]Beijing Academy of Quantum Information Sciences, Beijing, China. [3]College of Chemistry and Molecular Engineering, Peking University, Beijing, China. [4]State Key Laboratory of Optoelectronic Materials and Technologies, School of Materials Science and Engineering, Sun Yat-sen University, Guangzhou, China. [5]Center for Advanced Quantum Studies and Department of Physics, Beijing Normal University, Beijing, China. [6]Key Laboratory of Magnetic Materials and Devices, Zhejiang Province Key Laboratory of Magnetic Materials and Application Technology, Ningbo Institute of Materials Technology and Engineering, Chinese Academy of Sciences, Ningbo, China. [7]Department of Physics and State Key Laboratory of Surface Physics, Fudan University, Shanghai, China. [8]Hefei National Laboratory, Hefei, China. [9]Key Laboratory for the Physics and Chemistry of Nanodevices, Peking University, Beijing, China. [10]These authors contributed equally: Shaomian Qi, Di Chen. ✉e-mail: chenjianhao@pku.edu.cn

has been demonstrated in graphene quantum ferromagnet/anti-ferromagnet[11,12], layered anti-ferromagnet CrI3[13,14], CrSBr[15] and MnPS3[16], but in-plane anisotropic properties of 2D magnets are much less explored. As electrically controlled magnetic anisotropy plays an important role in spintronics[17,18], in-situ electrical modulation of anisotropy in spin transport could also enable vital applications like storage and logic operation based on 2D magnons in van der Waals magnets. However, such in-situ electrical modulation has yet to be achieved.

Here, we report the realization of electrically tunable anisotropy of diffusive magnon transport from isotropic to an anisotropy ratio of over 2,500,000% in van der Waals antiferromagnetic insulator $CrPS_4$. Two-dimensional anisotropic spin Seebeck effect is found to be central to the electrical tunability of the magnonic spin transport anisotropy. Based on such widely tunable anisotropy, a new type of ROM is demonstrated that is built on van der Waals magnons and have multi-bit parallel output.

## Results

$CrPS_4$ is an antiferromagnetic insulator that belongs to the family of ternary van der Waals magnetic materials MAX_n (here M = Mn, Fe, Ni, Cr, Co, A = P, Si, Ge, and X = S, Se, Te, n = 3, 4)[19]. $CrPS_4$ has monoclinic symmetry with space group $C2$. The Cr atoms form nearly rectangles in a layer, where the distance between neighboring Cr atoms along the $x$ axis (defined as <100> direction) is ~0.543 nm while that along the $y$

axis (defined as <010> direction) is ~0.355 nm or 0.371 nm[20,21]. The Néel temperature of $CrPS_4$ is around 38 K for bulk crystals[22] and the anti-ferromagnetism originates from transition metal ions $Cr^{3+}$, where each $Cr^{3+}$ ion carrying $2.8\mu_B$ of magnetic moments[21,23]. As shown in Fig. 1a, the ground magnetic state of $CrPS_4$ is a layered A-type antiferromagnet where intralayer spins have mostly out-of-plane Ising-like ferromagnetic alignment, while interlayer spins align antiferromagnetically[21]. The easy magnetization axis as determined via neutron powder diffraction is almost aligned to the $z$ axis with a small tilting angle of ~9.5° along the $x$ axis[22]. The interlayer antiferromagnetic exchange coupling is up to 18.5 times weaker than intralayer exchange coupling along the <010> axis direction (Fig. 1b), thus the magnetic system is rather two-dimensional. Furthermore, $CrPS_4$ crystals are air stable and magnetically ordered down to the monolayer limit[23], making $CrPS_4$ rather attractive candidate for applications in van der Waals spintronics.

The low structural symmetry of $CrPS_4$ not only results in out-of-plane magnetic anisotropy, but also creates marked in-plane magnetic exchange anisotropy. Figure 1b shows the ferromagnetic exchange coupling energy for three pairs of nearest adjacent Cr atoms in the $CrPS_4$ layer determined via neutron scattering[21], which is 0.51 meV along the <100> direction, as well as 2.09 meV and 2.96 meV along the <010> direction (Fig. 1b). Here the two values along the <010> axis is a result of the alternating Cr-Cr distance along this direction[21]. In order to probe the effect of structural and magnetic in-plane anisotropy on magnon generation and modulation, two magnon valve devices are

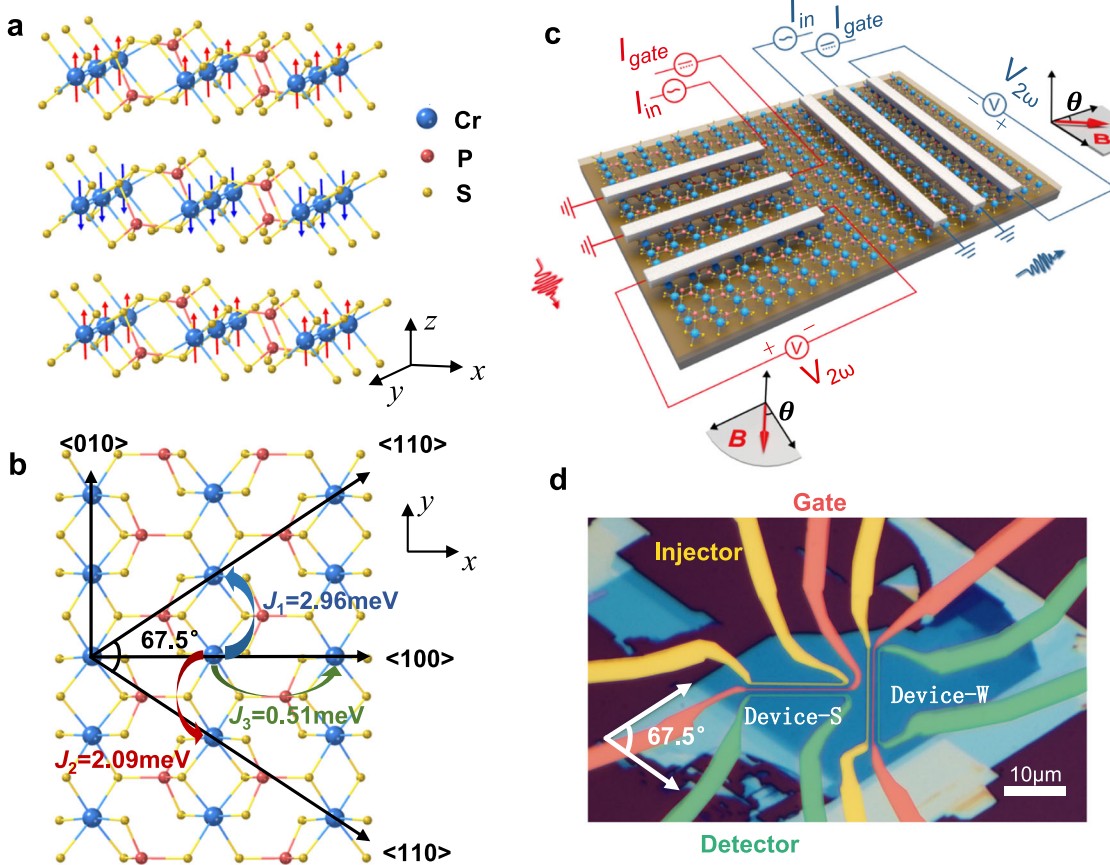

**Fig. 1 | $CrPS_4$ structure and anisotropic magnon transport measurement geometry. a** Crystal and spin structures of anti-ferromagnetic insulator $CrPS_4$. **b** In-plane atom arrangement indicating anisotropic magnetic exchange energies along the <100> and <010> direction, as well as the preferred cleavage crystallographic direction <110>. **c** Artistic schematics of the anisotropic magnon valve devices with external circuits and direction of the applied in-plane magnetic field. Specifically,

$I_{in}$: AC injection current; $I_{gate}$: DC gate current; $V_{2\omega}$: the second harmonic thermal magnon inverse spin Hall signal; $\theta$: the angle of the in-plane magnetic field with respect to the direction perpendicular to the Pt electrodes. **d** Optical micrograph of the devices, in which the Pt electrodes of Device-S (Device-W) are perpendicular to the <010>(<100>) direction, respectively. The injectors, gates and detectors are colored with yellow, red and green, respectively.

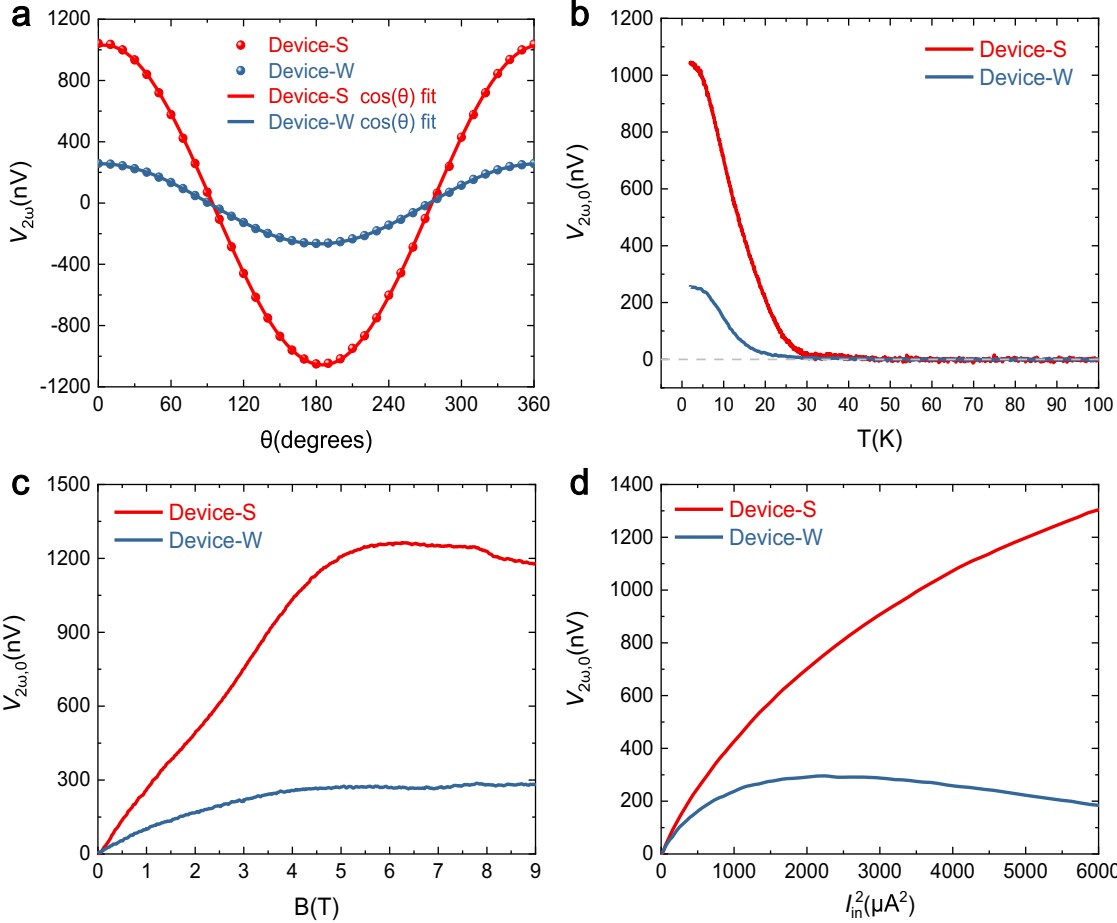

**Fig. 2 | Anisotropic second harmonic magnon signal $V_{2\omega}$. a** $V_{2\omega}$ vs. angle $\theta$ between the external **B** field and the direction perpendicular to respective Pt electrodes for Device-S and Device-W. The solid lines are fits to a cosine function. Both magnon valves have the same channel length of 750 nm and are excited with $I_{in} = 60 \mu A$. **b** $V_{2\omega}$ vs. $T$ at $\theta = 0$ ($V_{2\omega,0}$ vs. $T$). **c** Magnetic field dependence of $V_{2\omega,0}$. **d** $V_{2\omega,0}$ vs. the square of the injection current $I_{in}^2$. Here, experimental data and fitting curves for Device-S (Device-W) are in red (blue), respectively.

fabricated with the magnon propagation direction along the <100> and <010> axis, respectively, as shown in Fig. 1c. The structure of any one of these magnon valves resembles the magnon valves with MnPS$_3$ channel[16], with the only difference in the channel materials. Note that MnPS$_3$ thin flakes are van der Waals anti-ferromagnetic insulators with isotropic in-plane magnetic exchange coupling. The magnon valves contain van der Waals magnetic channel material and three Platinum (Pt) wires including an injector, a gate electrode and a detector. An AC current $I_{in}$ with frequency $\omega$ is applied through the injector to thermally generate diffusive magnons in the channel; a DC current $I_{gate}$ is applied through the gate electrode between injector and detector to tune the signal at the detector; then the modulated second-harmonic nonlocal inverse spin Hall voltage $V_{2\omega}$ is measured from the detector[24,25]. The energy band gap of the CrPS$_4$ channel is around 1.4 eV [26], which is large enough to prevent electrical conduction in the channel (see Supplementary Information S1). In such configuration, an in-plane magnetic field is needed to tilt the out-of-plane spins for getting an in-plane component according to the orthogonal rule of inverse spin Hall effect (ISHE)[24]. Note that the first-harmonic nonlocal inverse spin Hall voltage $V_{1\omega}$ is not observed in our experiment. It has been previously reported that the second-harmonic signal $V_{2\omega}$ increases with decreasing temperature, while $V_{1\omega}$ shows the opposite trend and significantly decreases with decreasing temperature for YIG[27]. If the same trend applies to other anti-ferromagnetic insulators such as CrPS$_4$, the first-harmonic signal in CrPS$_4$ is likely too weak to be detected in our case, since our experiments are carried out at rather

low temperatures, limited by the Néel temperature of 38 K for the crystal.

To study the anisotropic magnon transport, it is necessary to fabricate the two magnon valve devices strictly along the <100> and <010> crystallographic directions. The achievement of such alignment is ensured by optically identifying the preferred cleavage direction <110>[26,28] before device fabrication, and by angled-dependent polarized Raman spectroscopy[29,30] after device fabrication (details in Supplementary Information S2). For a magnon valve device with Pt wires parallel to the <100> direction, it detects magnon propagation along the <010> direction, where stronger exchange coupling exists. Such a device is labeled as Devices-S ("S" stands for stronger exchange coupling). Conversely, a magnon valve device with Pt wires parallel to the <010> direction would detect magnon transport along the <100> direction, and is labeled as Device-W ("W" stands for weaker exchange coupling). After fabrication of CrPS$_4$ magnon valve Device-S and Device-W with the same device dimensions and on the same thin flake, we first characterize their magnon transport properties without gating. Figure 2a shows the SHM signal $V_{2\omega}$ vs. $\theta$ of Device-S and Device-W at $T = 2$ K and $B = 4$ T. Here $\theta$ is the angle of the in-plane magnetic field **B** with respect to the perpendicular direction of the respective detector Pt electrodes. An AC injection current $I_{in}$ of root mean square value 60 μA is applied with a frequency of 17.77 Hz. Owing to the orthogonal rule of ISHE at the detector, $V_{2\omega}$ has dependence, consistent with thermally generated magnons[25]. It is experimentally found that Device-S has stronger signals than Device-W with identical injection current

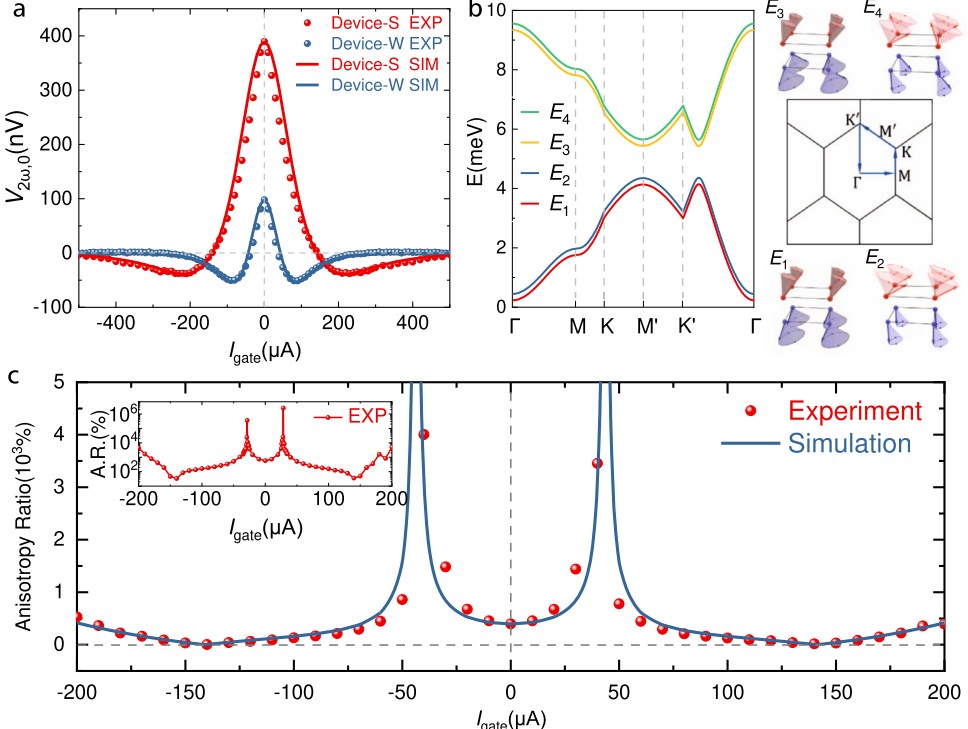

**Fig. 3 | Electrical tunable anisotropic magnon transport and theoretical simulation. a** Experimental and simulated $V_{2\omega,0}$ vs. $I_{gate}$ curves at $\mathbf{B} = 4T$, $I_{in} = 60\,\mu A$ and $T = 2$ K. Red and blue points show magnon transport signals taken along the <010> (Device-S) and <100> (Device-W) directions of the CrPS$_4$ crystal, respectively. Red and blue solid curves are respective simulations for Device-S and Device-W. **b** Left panel: The dispersions relation of spin waves along high symmetry directions of the CrPS$_4$ crystal under an in-plane magnetic field of 4 T. Upper right and lower right panels: Cartoons schematics of the spin precession of each mode. Middle right panel: Illustration of the high symmetry directions in the Brillouin zone of Cr monoclinic lattice in CrPS$_4$. **c** Experimental data (dots) and simulation (solid line) of the anisotropy ratio $|V_{2\omega,0}^S/V_{2\omega,0}^W|$ vs. $I_{gate}$. Inset shows the experimental $|V_{2\omega,0}^S/V_{2\omega,0}^W|$ vs. $I_{gate}$ from another device.

and device geometry. Next, we fixed the **B** field at $\theta = 0$ and studied $V_{2\omega}(\theta = 0)$ vs. $T$. We abbreviated $V_{2\omega}(\theta = 0)$ as $V_{2\omega,0}$ in following description for simplicity. As shown in Fig. 2b, both of the SHM signals for the two devices decrease with increasing temperature, and vanish at around 25 K.

We also studied the **B** field dependent $V_{2\omega,0}$ of the two devices (Fig. 2c). The signals first increase with the **B** field and then saturate at 5 T for Device-S and 4 T for Device-W. The increase of signals at lower **B** field corresponds to the moment gradually rotating to the in-plane direction with increasing **B**. The saturation of signals at higher **B** field could be due to a suppression of the magnon diffusion length or magnon population by external magnetic field[31–33]. The anomalous kink of $V_{2\omega,0}$ around 8 T in Device-S may come from a spin saturation transition, which is near the critical spin-flip magnetic field of 8.5 T for CrPS$_4$ [22]. Figure 2d shows $V_{2\omega,0}$ versus $I_{in}^2$ for Device-S and Device-W. With increasing injection power, both $V_{2\omega,0}$ deviate from the linear dependence of $I_{in}^2$, which may be caused by the increase of temperature at the detector electrodes[32]. As can be seen from Fig. 2a–d, the overall SHM signal $V_{2\omega}$ is higher for Device-S compared to Device-W, no matter how the parameters $T$, **B** or $I_{in}$ varied. This intrinsic anisotropic response is consistently observed in all devices (>10 pairs, see Supplementary Information S3 for data from more devices) we studied, which manifests the profound effects of structural anisotropy to the magnonic spin transport properties of CrPS$_4$, and is already useful for applications as-is. In the following, we shall show that such anisotropy can be greatly amplified due to the highly tunable nature of 2D magnons, which could enable critical applications.

The application of DC current $I_{gate}$ through the gate electrodes in Device-S and Device-W is found to have strong and anisotropic effects to the SHM signal $V_{2\omega,0}$ in respective devices. Figure 3a shows the $V_{2\omega,0}(I_{gate})$ curves of the two magnon valves with channel lengths of

1.5μm, $I_{in} = 60\mu A$ at $T = 2$ K and **B** = 4 T with $\theta = 0$. Figure S4 in Supplementary Information shows the normalized $V_{2\omega,0}(I_{gate})/V_{2\omega,0}(I_{gate} = 0)$ curves where the differences in the functional dependence of $V_{2\omega,0}$ on $I_{gate}$ for the two devices are more clearly shown. The solid lines in Fig. 3a are simulations based on our theoretical model and will be discussed later.

The features of the $V_{2\omega,0}(I_{gate})$ curves from Device-S and Device-W can be summarized as the following: (1) The symmetric tuning response $V_{2\omega,0}(+I_{gate}) = V_{2\omega,0}(-I_{gate})$ implies a spin Seebeck nature of this gating effect, which is confirmed by magnetic field angle dependent data (see Supplementary Information S5) and is consistent with previous study on van der Waals magnon valves[16]; (2) Each $V_{2\omega,0}(I_{gate})$ curve has one zero point and has negative values after the zero points, also consistent with magnon valve behavior in MnPS$_3$[16]; (3) Strong anisotropy is found in the $V_{2\omega,0}(I_{gate})$ curves of Device-S and Device-W, specifically, the zero point for Device-S is $I_0^S = 140\mu A$ and the signal vanishes at around 500μA, while for Device-W we have $I_0^W = 40\mu A$ and the signal vanishes at around 260μA; (4) The maximum negative value of $V_{2\omega,0}$ for Device-S is about 10% of $V_{2\omega,0}(I_{gate} = 0)$, while such ratio in Device-W is 55% (see Supplementary Fig. S4). Among these features, the existence of different zero points of the $V_{2\omega,0}(I_{gate})$ curves for Device-W and Device-S could be particularly useful for anisotropic magnon operations. Additional experiments have also been carried out to rule out the possibility of a phonon carried thermal effect or an anomalous Nernst effect in our devices. In particular, the local spin Seebeck signal of the injector electrode could not be tuned to inverse as a function of $I_{gate}$, which shows that the magnon diffusive process and anisotropic magnetic exchange interactions are vital in producing the highly tunable anisotropic nonlocal signal. Furthermore, we measured the nonlocal second harmonic signal with an applied magnetic field of up to 9 T rotated in the *x-z* plane. The signal is almost zero when the magnetic

field is along the $z$ axis, which indicates the absence of an anomalous Nernst effect (details in Supplementary Information S6–S7).

Dots in Fig. 3c shows the experimental data of the anisotropy ratio $|V_{2\omega,0}^{S}/V_{2\omega,0}^{W}|$ vs. $I_{gate}$, where it can be electrically tuned continuously from 100% (completely isotropic) to about 4000% (extremely anisotropic) by adjusting the $I_{gate}$ from 140μA to 40μA, which is much larger than the magnetic field tuned magnon transport anisotropy of ~200% reported in $\alpha$-Fe$_2$O$_3$ thin film[34] and crystalline anisotropy induced magnon transport anisotropy of ~150% reported in MgAl$_{0.5}$Fe$_{1.5}$O$_4$ thin film[35]. As a matter of fact, such extremely anisotropic magnon transport is bound by the discrete $I_{gate}$ values taken during our measurement (i.e. $I_0^W$ is not exactly at 40μA). By taking $I_{gate}$ points exactly at the zero crossing of the $V_{2\omega,0}(I_{gate})$ curves for Device-W, an anisotropy ratio of over 2,500,000% is achieved in another CrPS$_4$ device, as shown in the inset of Fig. 3c. Such exceptionally large anisotropic ratio is limited only by the noise floor (<1nV) of our experimental apparatus, and could be even larger (tending to infinity) as shown by the simulation curve based on our theoretical model (solid line in Fig. 3c).

To understand the nature of this electrically tunable anisotropic magnon transport, we propose a two-dimensional localized spin model with easy-axis single-ion anisotropy and in-plane exchange anisotropy to describe the antiferromagnetic insulator CrPS$_4$. Finite element analysis[36] and three-dimensional spin model were performed, and the effect of thermal gradient and spin Seebeck coefficient along the $z$ direction is estimated to be small (details in Supplementary Information S10–S11). The 2D nature of the magnon transport in CrPS$_4$ is warranted by the weak interlayer exchange interactions[21]. Since CrPS$_4$ is a layered A-type antiferromagnet, i.e., each Cr atomic layer is an Ising ferromagnet with interlayer antiferromagnetic coupling, we consider a bilayer spin model under a transverse magnetic field to match with the experimental conditions of the magnon valve devices. The spin Hamiltonian of such bilayer model is:[16,21]

$$H = H_1 + H_2 + H_{1,2} \tag{1a}$$

$$H_1 = \sum_{\boldsymbol{j}} \sum_{m=1,2,3,4} J_{\boldsymbol{a_m}} \boldsymbol{S}_{1\boldsymbol{j}}^{A} \cdot \boldsymbol{S}_{1\boldsymbol{j}+\boldsymbol{a_m}}^{B} - D \sum_{\boldsymbol{j}} \left[ \left( S_{1\boldsymbol{j}}^{A,z} \right)^2 + \left( S_{1\boldsymbol{j}}^{B,z} \right)^2 \right] - h \sum_{\boldsymbol{j}} \left[ S_{1\boldsymbol{j}}^{A,y} + S_{1\boldsymbol{j}}^{B,y} \right]$$

$$H_2 = \sum_{\boldsymbol{j}} \sum_{m=1,2,3,4} J_{\boldsymbol{a_m}} \boldsymbol{S}_{2\boldsymbol{j}}^{A} \cdot \boldsymbol{S}_{2\boldsymbol{j}+\boldsymbol{a_m}}^{B} - D \sum_{\boldsymbol{j}} \left[ \left( S_{2\boldsymbol{j}}^{A,z} \right)^2 + \left( S_{2\boldsymbol{j}}^{B,z} \right)^2 \right] - h \sum_{\boldsymbol{j}} \left[ S_{2\boldsymbol{j}}^{A,y} + S_{2\boldsymbol{j}}^{B,y} \right] \tag{1b, c}$$

$$H_{1,2} = J_c \left[ \sum_{\boldsymbol{j}} \boldsymbol{S}_{1\boldsymbol{j}}^{A} \cdot \boldsymbol{S}_{2\boldsymbol{j}}^{A} + \boldsymbol{S}_{1\boldsymbol{j}}^{B} \cdot \boldsymbol{S}_{2\boldsymbol{j}}^{B} \right] \tag{1d}$$

where $J_{\boldsymbol{a_1}} = -2.96meV$, $J_{\boldsymbol{a_2}} = -2.09meV$, $J_{\boldsymbol{a_3}} = J_{\boldsymbol{a_4}} = -0.51meV$ are the four intralayer exchange coupling of nearest neighbor Cr atoms, $D = 0.0058\,meV$ is the single-ion anisotropy energy, $h$ is the external magnetic field, $J_c = 0.16meV$ is the interlayer exchange coupling of nearest neighbor Cr atoms. The values of the above parameters are taken from a neutron scattering experiment[21]. $H_1$ and $H_2$ are the spin Hamiltonians of the first and second Cr layer, respectively; $H_{1,2}$ is the interlayer exchange energy of the bilayer CrPS$_4$; thus, $H$ is the total Hamiltonian of the bilayer spin system.

In Fig. 3b, the left panel shows the calculated magnon band structure of a bilayer CrPS$_4$ under an in-plane magnetic field of 4 T based on Eq. 1a; the upper right and lower right panels show the schematics of the corresponding spin wave modes; the middle right panel shows the first Brillouin zone of the magnon energy band (more

details in Supplementary Information S8). Apparent anisotropy along $\Gamma M$(i.e., <100>) and $\Gamma K'$(i.e., <010>) directions is revealed on the magnon band structure. We found that such anisotropy not only affects the 2D magnon group velocity along different crystallographic directions, but also influences the electrical tunability of such 2D magnons. Specifically, the inverse spin Hall voltage $V_{ISHE}$ generated by thermal magnons at the detector electrode is proportional to:[37–40]

$$\boldsymbol{V}_{ISHE}\left(I_{in}(t), I_{gate}\right) \propto g_{mix}\hat{\boldsymbol{n}} \cdot \boldsymbol{S}(T) \cdot \boldsymbol{\nabla}T \tag{2}$$

where $g_{mix}$ is the real part of the effective spin mixing conductance at the CrPS$_4$/Pt interface. $\hat{\boldsymbol{n}} = \hat{\boldsymbol{x}}, \hat{\boldsymbol{y}}$ are unitary vectors corresponding magnon propagation directions in Device-W and Device-S, respectively. $\boldsymbol{S}(T)$ is the spin Seebeck coefficient tensor and $\boldsymbol{\nabla}T$ is the temperature gradient within the 2D film.

$\boldsymbol{S}(T)$ in CrPS$_4$ under in-plane magnetic field can be derived based on a semi-classical Boltzmann transport theory of 2D magnons (details at Supplementary Information S9):[41]

$$\boldsymbol{S}(T) = \frac{\hbar}{(2\pi)^2 k_B T^2} \sin\psi \int_{BZ} dk_x dk_y \sum_{i=1}^{4} \boldsymbol{v}_i(\boldsymbol{k}) \cosh\delta_i \boldsymbol{v}_i(\boldsymbol{k}) \frac{e^{\hbar\omega_i/k_B T}\hbar\omega_i}{\eta_{i,k}\left(e^{\hbar\omega_i/k_B T} - 1\right)^2} \tag{3}$$

where $\psi$ is the canting angle of the spins from $z$ axis under finite in-plane magnetic field, $\sin\psi\cosh\delta_i$, $\eta_{i,k} = 1/\tau_{i,k}$, $\omega_i(\boldsymbol{k})$ and $\boldsymbol{v}_i(\boldsymbol{k})$ are the in-plane spin polarization of the magnon density of states, the magnon relaxation rate, the magnon dispersion relation, and the magnon group velocity, respectively, for the $i^{th}$ magnon branch at magnon momentum $\boldsymbol{k}$ (details in Supplementary Information S9).

Considering the Joule heating from current applied in the injector and the gate, an average temperature increase in the device can be expressed as $\Delta T = c_1 I_{in}^2 + c_2 I_{gate}^2$, where $c_1$ and $c_2$ are factors accounting for different heating effects induced by the injector and the gate, involving the resistance of the Pt bar, the specific heat of CrPS$_4$ and various interfacial heat resistance and channel heat conductance in the device. Consequently, the magnon temperature can be written as $T = 2K + c_1 I_{in}^2 + c_2 I_{gate}^2$, where 2K is the base temperature in our experiment. The temporal dependence of $V_{ISHE}$ comes purely from the time variation of $I_{in}^2(t) \propto \sin^2(\omega t)$. By substituting $T$ in $\boldsymbol{S}(T)$ with $2K + c_1 I_{in}^2 + c_2 I_{gate}^2$, we can use the following equation to fit our $V_{2\omega,0}^S$ and $V_{2\omega,0}^W$ data (labeled as $V_{2\omega,0}^{S,W}$):

$$\begin{aligned} V_{2\omega,0}^{S,W} &= C^{S,W*}\left[\hat{\boldsymbol{n}}\,\boldsymbol{S}(T)\cdot\boldsymbol{\nabla}T\right]_{2\omega} \\ &= C^{S,W*}\left[S_n\left(2K + c_1^{S,W}I_{in}^2 + c_2^{S,W}I_{gate}^2\right)*\left(c_1^{S,W}I_{in}^2 + c_2^{S,W}I_{gate}^2\right)\right]_{2\omega} \end{aligned} \tag{4}$$

where $V_{2\omega,0}^S$ ($V_{2\omega,0}^W$) is the SHM signal of Device-S (Device-W), $C^{S,W}$ is a global parameter containing $g_{mix}$, $S_n$ is the spin Seebeck coefficient component along $\hat{\boldsymbol{n}}$ direction, $c_1^{S,W}$ and $c_2^{S,W}$ are heating efficiencies of the injector and gate along the two directions, respectively, and $[\ldots]_{2\omega} = \frac{\omega}{\pi}\int_{-\frac{\pi}{\omega}}^{\frac{\pi}{\omega}}\cos(2\omega t)*[\ldots]dt$ means taking the second harmonic component.

Based on the above model, we simulated the gate dependence of $V_{2\omega,0}^S$ for Device-S and $V_{2\omega,0}^W$ for Device-W as shown in the solid lines in Fig. 3a; the simulated anisotropic ratio $|V_{2\omega,0}^S/V_{2\omega,0}^W|$ versus $I_{gate}$ is also plotted as a solid line in Fig. 3c. All the simulated curves agree well with our experimental data with a set of fitted global parameters $C^S = 0.527 \times 10^{-26}$V · s/ℏ, $c_1^S = 1.7 \times 10^{-4}$K/μA$^2$, $c_2^S = 1.2 \times 10^{-4}$K/μA$^2$, $C^W = 0.484 \times 10^{-26}$V · s/ℏ, $c_1^W = 4.9 \times 10^{-4}$K/μA$^2$, $c_2^W = 3.8 \times 10^{-4}$K/μA$^2$. The same set of global parameters can also produce $V_{2\omega,0}^{S,W}$ vs. $I_{gate}$ curves at different **B** fields that agree well with experiment (see Supplementary Fig. S12), indicating that our model captured the main physical process in the magnon generation and manipulation in CrPS$_4$ magnon valves. According to the model, the key to the realization of electrically tunable magnon transport anisotropy in CrPS$_4$ comes from the anisotropic spin

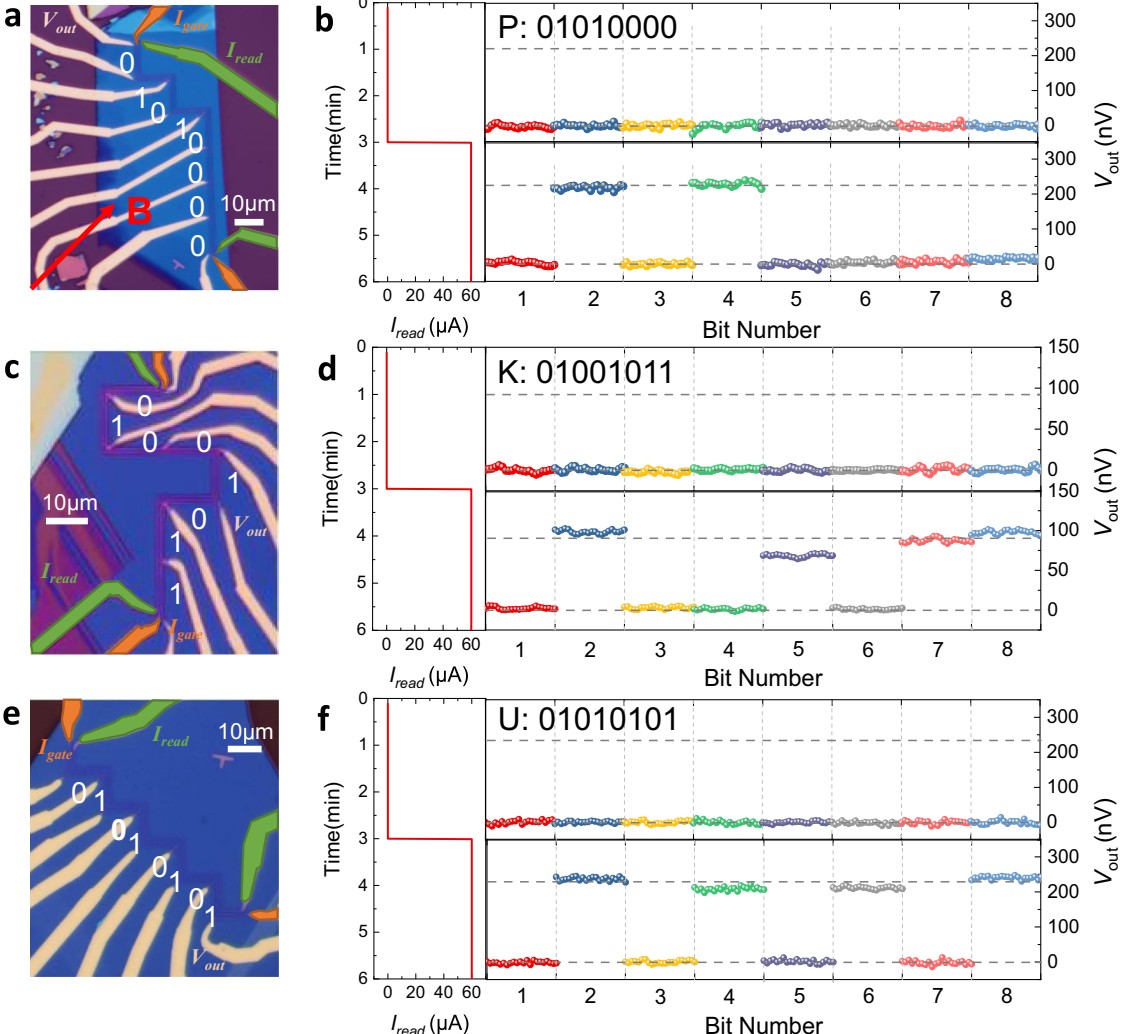

**Fig. 4 | Anisotropic magnon read-only memory (ROM).** Left panels **a**, **c**, **e**: Optical image of each CrPS$_4$ anisotropic magnon read-only memory unit recording the letter "PKU" in ASCII codes. The read electrodes and gate electrodes are marked with green and orange color, respectively, while the output electrodes are of their original color. Right panels **b**, **d**, **f**: operation of the magnon ROM with $I_{read}$ = 0µA (not reading) and 60µA (reading), respectively. When in the reading state, each magnon ROM unit generates an 8-bit series that represent a capital letter according to the ASCII codes.

Seebeck coefficient tensor **S** shown in Eq. 3. There are three specific implications of such **S**. First, $S_{yy}$ is larger than $S_{xx}$ under the same excitation with zero $I_{gate}$, resulting in stronger signal for Device-S than Device-W before gating (i.e., $I_{gate}$ = 0). Here $S_{yy}$ and $S_{xx}$ is the spin Seebeck coefficient matrix element along the <010> direction and <100> direction, respectively (see Figs. 2 and 3a, Supplementary Figs. S3 and S5). Second, $I_0^S$ is much larger than $I_0^W$. Here $I_0^S$ and $I_0^W$ are the zero points at the $V_{2\omega,0}^S(I_{gate})$ and $V_{2\omega,0}^W(I_{gate})$ curves, respectively (see Fig. 3a and Supplementary Figs. S4, S12a–12d). Third, the negative maximum of $V_{2\omega,0}^W(I_{gate})$ curves are proportionally much larger than that of $V_{2\omega,0}^S(I_{gate})$ curves (see Supplementary Figs. S4 and S12a–12d). Meanwhile, even though anisotropic $\nabla T$ (i.e., $c_1^S \neq c_1^W$ and $c_2^S \neq c_2^W$) also has an influence on the anisotropic gate tuning effect, simulation using isotropic $\nabla T$ with only exchange anisotropy generate the same key features observed experimentally (details in Supplementary Information S13 and Fig. S10.2).

Based on the highly tunable anisotropic behavior in CrPS$_4$ magnon valves, a novel type of multi-bit van der Waals magnon ROMs with parallel readout can be constructed. Such van der Waals magnon ROM can be fabricated with three parallel polylines of Pt wires deposited on CrPS$_4$ single crystal with 90° angle at each turn. Such polylines are orientated so that each segment of the polyline is

perpendicular to the <100> or <010> direction of the CrPS$_4$ crystal so that magnon signal along these two directions can be excited, modulated and detected. The first polyline has two electrical leads, one at each end of this polyline, so that an global injection current $I_{in}$ (in the setting of the magnon ROM, this current is called the reading current $I_{read}$) can run through all the segments simultaneously; similarly, the middle polyline also has two leads to allow for the application of a global gate current $I_{gate}$; the third polyline, on the other hand, has a pair of electrical leads on every segment of the polyline which allows for parallel output of the SHM signal from each Pt wire segment. By applying a magnetic field $B$ = 4 T along the $\theta$ = 45° direction from the <100> direction and applying a global DC gate current $I_{gate} = I_0^W$, the ratio of the SHM signals from each orthogonal segments of the detector polyline could reach to a maximum value since the thermal magnon transport is completely suppressed along the <100> direction. Note that there is a large range of $I_{gate}$ that could be used other than a specific $I_0^W$, since $V_{2\omega,0}^W(I_{gate})$ vanishes at $I_{gate} \gtrsim 260\mu A$, while $V_{2\omega,0}^S(I_{gate})$ maintains to $I_{gate}$ over 400µA (Fig. 3a). Thus, in the configuration of the magnon ROMs, with $I_{gate} = I_0^W$ or $260\mu A \lesssim I_{gate} \lesssim 400\mu A$, segments detecting magnons along the <010> direction would generate "1" and segments detecting magnons along the <100> direction would generate "0" when $I_{read}$ is

turned on. By building these segments according to specific coding sequence, a parallel magnon ROM can be realized.

The left panels of Fig. 4 display three optical images of the CrPS$_4$ magnon ROM units recording the capital letters "PKU" abbreviated for Peking University. According to the ASCII binary codes, "01010000", "01001011" and "01010101" represents "P", "K" and "U", respectively. Three parallel polylines of Pt wires with particular segment orientations deposited on CrPS$_4$ flakes constitute the three ROM units. In the demonstration for the 24-bit ROM, every eight binary bits share the same reading electrode (the first Pt polyline) and the same gate electrode (the middle Pt polyline), representing one letter. With a proper $I_{gate}$, eight bits can be read out in parallel from pairs of electrical leads on every segment of the output electrode (the third Pt polyline) at one time when an $I_{read} = 60\mu A$ is applied to the injector circuit. The right panels of Fig. 4 show the respective readout of such ROM units with and without a reading current. When $I_{read} = 0\mu A$, all the output voltages are zero; when $I_{read} = 60\mu A$, each ROM unit generates an 8-bit serial code representing "PKU" together. Since the information is inscribed in the anisotropy of the magnetic lattice of CrPS$_4$, the stability of CrPS$_4$ as well as its intrinsic anisotropy guarantees high durability and stability of such magnon ROM. Experimentally, the CrPS$_4$ anisotropic magnon ROM shows rather stable readout states in a time scale of at least 3 min (each section in Fig. 4b has a duration of 3 min). The reading process has been repeated for 100 cycles with little fluctuation of the output voltages, indicating outstanding reliability of the memory device. A CrPS$_4$ magnon valve device is also tested to show nearly identical readouts right after fabrication and about 9 months after fabrication (see Supplementary Fig. S14). Such stable magnon ROM provides a foundation for information storage naturally compatible with diffusive magnon based arithmetic and logic circuits in the near future.

## Discussion

In summary, strong and electrically tunable diffusive magnon transport anisotropy is discovered in van der Waals antiferromagnetic insulator CrPS$_4$, in which the ratio of the second harmonic thermal magnon signals along the two in-plane crystal axes can be tuned from isotropic to over 2,500,000% through the application of a modest DC electric gate current. The experimentally measured strong magnon transport anisotropy arises from the anisotropic spin Seebeck coefficient tensor and agrees well with our 2D magnon model. A novel multibit parallel ROM is demonstrated utilizing this highly tunable anisotropic magnon transport, unveiling the potential of van der Waals magnon in future information storage technology.

## Methods

### Device fabrication and sample characterization

CrPS$_4$ flakes are mechanically exfoliated from bulk crystals and deposited on 300 nm SiO$_2$/Si substrates. Thickness of the CrPS$_4$ flakes used in our study ranges from 2 to 30 nm. The injector, gate and detector electrodes in the magnon valves are fabricated with standard electron-beam lithography, platinum deposition and lift-off processes. Platinum is deposited in a magnetron sputtering system, and the width of the wires is ~250 nm with a thickness of 9 nm. The parallel Pt wires are 750 nm apart from each other. Afterwards, 5 nm of titanium and 80 nm of gold are patterned to contact the platinum wires. More than 10 pairs of anisotropic magnon valve devices were made and studied. Apart from the memory devices, data from five anisotropic magnon valve devices (device 1–device 5) were presented. Data shown in the main text were obtained from device 1 (Fig. 2), device 2 (Fig. 3) and device 3 (inset of Fig. 3d), and the results for other devices are presented in the Supplementary Information.

### Nonlocal magnon transport measurement

The magnon transport measurement in CrPS$_4$ is done in a Physical Properties Measurement System (PPMS) with low-frequency lock-in

amplifier technique. The injection AC current (17.77 Hz) in the range from 0μA to 100μA is provided by lock-in amplifier (NF LI5640) with a 10KΩ resistor. Lock-in amplifiers (NF LI5640 and Stanford Research SR830) are used to probe the nonlocal voltages. Low noise voltage preamplifiers (NF LI75A) are also used. A voltage source (Keithley 2400) with a 100KΩ resistor is used to provide the DC current (0 μA- 500 μA) to modulate the nonlocal signal. The temperature of the measurement in PPMS ranges from 2 to 300 K, the applied magnetic field is parallel to our sample plane and the maximum field is 14 T.

## Data availability

Source data are provided with this paper. Data for figures that support the current study are available at https://doi.org/10.7910/DVN/QQG2HX.

## Code availability

Numerical simulations were performed with Matlab. Source codes in Matlab file format are available from the corresponding authors upon reasonable request.

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

## Acknowledgements

This project has been supported by the National Basic Research Program of China (Grant Nos. 2019YFA0308402 & 2018YFA0305604 (J.-H.C.), 2019YFA0308401 (R.S.), 2019YFA0308404 (S.W.), 2021YFE 0194200(P.Y.), 2015CB921100 (X.C.X.), 2021YFE0194200(X.C.X. and J.-H.C.)), the National Natural Science Foundation of China (NSFC Grant Nos. 11934001, 92265106 & 11921005 (J.-H.C.), 11534001 (X.C.X.)), Beijing Municipal Natural Science Foundation (Grant No. JQ20002 (J.-H.C.)), Natural Science Foundation of Guangdong Province No. 2020A1515110821 (P.Y.), and Guangzhou Science and Technology Project No.202102020126 (P.Y.). J.-H.C. acknowledge the technical supports from Peking Nanofab.

## Author contributions

J.-H.C. conceived the idea that directed the experiment; S.Q. fabricated the devices and performed most of the measurements; B.L., L.J., Y.S., J.L., and G.C. aided in transport measurement; M.H. aided in device fabrication and AFM measurement; B.L. and D.C. performed Raman spectroscopy and Raman data analysis; K.C., S.Q., J.L., X.C.X., R.S., and J.X. provided theoretical analysis; K.C. and B.L. performed simulations according to theoretical analysis; P.Y. grew high quality CrPS$_4$ bulk crystals; Y.L. and H.W. performed thermal analysis; D.C. and J.-H.C. wrote the manuscript; all authors commented and modified the manuscript.

## Competing interests

The authors declare no competing interests.
