## [Peer Review File · Nature Communications]

Giant electrically tunable magnon transport anisotropy in a van der Waals antiferromagnetic insulatorReviewers' Comments:

Reviewer #1:

Remarks to the Author:

In this paper, the authors reported non-local spin Seebeck effects in anisotropic van der Waals 2D magnets CrPS₄. They analyzed the anisotropy of the non-local second harmonic voltage with respect to the direction of the transport direction. The result shows that the anisotropy can be modulated significantly by applying a current to a thin wire placed between the heater wire and the Pt wire (the spin-current detector) on the CrPS₄.

The phenomenon is interesting, and the paper can provide an important piece of information for spintronics and 2D material science researchers after the authors rewrite the manuscript with careful analysis and description. I thus can recommend the publication of this paper after the author has made the following revisions.

(major points)

1. It is necessary to clarify where and how the heat flows in the sample.

Please remember that the driving force for Seebeck and spin Seebeck effects is temperature gradient (not a heat flow).

In general, in samples with in-plane thermal conductivity much greater than out-of-plane thermal conductivity, strong out-of-plane temperature gradient should be formed when a part of the surface is heated, while in-plane temperature gradient is much smaller. Therefore, in spite of the small SSE coefficient for the out-of-plane direction, the out-of-plane effect, such as SSEs, should be considered in a thick CrPS₄ film. Also, the absence of the $V_1\Omega$ signal suggests a minor role of the in-plane conduction.

Similar physics can be found in the non-local SSEs in YIG films (for example, Phys. Rev. B 96, 104441 2017), in which, in spite of the in-plane SSE configuration, out of plane temperature gradient and resultant spin chemical potential distribution plays a role.

2. Given the above considerations, Equation 2 should be reconsidered carefully.

3. In the abstract: 2500000% is oversold and misleading. It sounds as if it were an on-off ratio. It is not fair unless it is clearly written that the data is measured by different devices on the same film.

(minor points)

4. Line 190: A brief summary of the contents of the SI S6-7 may be added to the main text for better readability.

5. Line 262: Please add an explanation on the expansion of Equation 4.

6. Page 14- The ROM part does not provide new physics but the ROM has not been analyzed for its engineering merits from application points of view: the ROM seems too slow and too large (it is difficult to confine heat current into small area). This part can be compressed.

Reviewer #2:

Remarks to the Author:

In this manuscript, the authors report anisotropy in diffusive magnon transport in van Der Waals (vdW) antiferromagnetic insulator CrPS₄. This was accomplished by performing nonlocal magnon transport measurement along different crystal axes. The authors demonstrate that anisotropy is tunable via a gate current, with an anisotropy ratio of up to 2500x achieved. Lastly, by utilizing the observed anisotropic magnon transport, the authors demonstrate a read-only memory (ROM) whose value is set by placement of read electrodes.

The experiment is well done; the results and their interpretations are convincing. The scientific claims are well supported, and methodology is sound. The work should be reproducible given the details provided in the manuscript.

However, I am not convinced that the impact of the result is sufficient to warrant publication in Nature Communications. In the introduction, the authors have cited recent interest in various types of anisotropy, which supposedly is to argue for the significance of magnon-transport anisotropy studied in this work. Indeed, electrical and optical anisotropy have various anticipated applications as discussed in the literature, such as directional memory (Ref. 1), polarization-sensitive photodetectors (Refs. 4, 5), novel polarizers/polarization sensors (Xia, Wang, Jia, Nat. Commun. 5, 4458 (2014)), to name a few. However, it is not clear that there is high-impact device applications envisioned that relies on anisotropic diffusive magnon-transport studied in this work. There is a statement "In particular, in-situ electrical modulation of anisotropy in spin transport, a vital functionality for future large-scale applications of van der Waals magnets, has yet to be achieved"; why electrical modulation of anisotropy is vital to large-scale applications of vdW magnets is neither explained nor supported by citation. ROM based on anisotropic magnon transport is interesting, but it does not have prospect for future practical applications. In particular, this ROM requires a large field (4 T) which is not practical. Lastly, the electrical tunability relies on heating coming from gate current, which is likely not fast and possibly highly temperature-dependent, hence less of an ideal control modality compared to electric field gating (if it could be used to control anisotropy).

Given the above consideration, I cannot recommend publication in a high impact journal for diverse audience such as Nature Communications. I would recommend publication in journals more specific to materials or nano science; for example, this manuscript would be suitable for NPJ Quantum Materials or NPJ 2D Materials and Applications.

In addition, here I also list a few other comments:

- when discussing magnons in 2D magnets (around line 61), it would be appropriate to also cite <https://www.nature.com/articles/s41586-022-05024-1>

- there is this statement in line 87-90: "Furthermore, ferromagnetism with out-of-plane spin orientation is found in monolayer, air-stable CrPS4 crystals, making CrPS4 more attractive candidate for applications in van der Waals spintronics"

This sentence is confusing. What is "more attractive" compared to? Perhaps the author simply meant to say CrPS4 is "attractive", or "rather attractive". Also, it's not clear what are the features that the authors consider to make this material attractive for spintronics.

- Why not replace Fig. 3c with the inset, which shows giant tunability of anisotropy?

- Line 279: what does "Specifically, 1" refer to?

- it would appropriate to include scale bars in the optical images showing the devices

Reviewer #1 (Remarks to the Author):

In this paper, the authors reported non-local spin Seebeck effects in anisotropic van der Waals 2D magnets CrPS4. They analyzed the anisotropy of the non-local second harmonic voltage with respect to the direction of the transport direction. The result shows that the anisotropy can be modulated significantly by applying a current to a thin wire placed between the heater wire and the Pt wire (the spin-current detector) on the CrPS4.

The phenomenon is interesting, and the paper can provide an important piece of information for spintronics and 2D material science researchers after the authors rewrite the manuscript with careful analysis and description. I thus can recommend the publication of this paper after the author has made the following revisions.

We are grateful that the reviewer pointed out the significance of our work and recommend the publication of our manuscript after revision. We have made careful analysis and modifications to our manuscript according to the comments of our reviewers and we believe we have successfully addressed all the questions and comments from the reviewers. Our point-by-point reply is listed below.

(major points)

1. It is necessary to clarify where and how the heat flows in the sample.

Please remember that the driving force for Seebeck and spin Seebeck effects is temperature gradient (not a heat flow).

In general, in samples with in-plane thermal conductivity much greater than out-of-plane thermal conductivity, strong out-of-plane temperature gradient should be formed when a part of the surface is heated, while in-plane temperature gradient is much smaller. Therefore, in spite of the small SSE coefficient for the out-of-plane direction, the out-of-plane effect, such as SSEs, should be considered in a thick CrPS4 film. Also, the absence of the $V_{1\Omega}$ signal suggests a minor role of the in-plane conduction.

Similar physics can be found in the non-local SSEs in YIG films (for example, Phys. Rev. B 96, 104441 2017), in which, in spite of the in-plane SSE configuration, out of plane temperature gradient and resultant spin chemical potential distribution plays a role.

We thank the reviewer for pointing out the necessary to clarify how the heat flows in the sample and how it affects the distribution of the temperature gradience in the sample in the in-plane and out-of-plane directions. Indeed, this is an important point that might help us better understand the excellent performance we measured from the CrPS₄ magnon devices experimentally.

According to the reviewer's comments, we perform additional finite element analyses of the temperature distribution and magnon chemical potential distribution for CrPS₄ magnon valve devices with crystal thickness of 30nm. We use the methods mentioned in Phys. Rev. B 96, 104441 (2017) as suggested by the reviewer.

The linear response relation of heat and spin transport in the bulk of a magnetic insulator reads:

$$\begin{pmatrix} \frac{2e}{\hbar} \mathbf{j}_m \\ \mathbf{j}_Q \end{pmatrix} = - \begin{pmatrix} \sigma_m & \mathbf{S}/T \\ \mathbf{S} & \kappa \end{pmatrix} \begin{pmatrix} \nabla \mu_m \\ \nabla T \end{pmatrix}$$

where \mathbf{j}_m is the magnon spin current, \mathbf{j}_Q the total heat current, μ_m the magnon chemical potential, T the temperature, σ_m the magnon spin conductivity, κ the total heat conductivity and \mathbf{S} the spin Seebeck coefficient. Combined with $\nabla \cdot \mathbf{j}_Q = \frac{j_c^2}{\sigma_{Pt}}$, and $\nabla \cdot \mathbf{j}_m = -\frac{\hbar \sigma_m}{2e \lambda_m^2} \mu_m$, the diffusion equations for spin and heat read:

$$\begin{aligned} \mathbf{S} \cdot \nabla^2 \mu_m + \kappa \cdot \nabla^2 T &= -\frac{j_c^2}{\sigma_{Pt}} \\ \sigma_m \cdot \nabla^2 \mu_m + \mathbf{S} \cdot \nabla \left(\frac{\nabla T}{T} \right) &= \frac{\sigma_m \cdot \mu_m}{\lambda_m^2} \end{aligned}$$

where j_c is the charge current density in the injector Pt electrode, σ_{Pt} is the electrical conductivity of the Pt electrode and λ_m the magnon spin diffusion length.

Since CrPS₄ is not as widely studied as YIG, some parameters are unavailable, let alone considering the anisotropy. For the parameters related to temperature gradient distribution analysis, there is no thermal conductivity data found for CrPS₄ in the literature. Here we use the in-plane thermal conductivity 6.3[W/(m*K)] for MnPS₃ as an rough estimate of the thermal conductivity for CrPS₄ along the <010> direction, and the through-plane thermal conductivity 1.1[W/(m*K)] for MnPS₃ as the through-plane thermal conductivity for CrPS₄. Based on our fitting parameters $c_1^S = 1.7 \times 10^{-4} \text{K}/\mu\text{A}^2$, $c_2^S = 1.2 \times 10^{-4} \text{K}/\mu\text{A}^2$ for Device-S (magnon transports along the <010> direction) and $c_1^W = 4.9 \times 10^{-4} \text{K}/\mu\text{A}^2$, $c_2^W = 3.8 \times 10^{-4} \text{K}/\mu\text{A}^2$ for Device-W (magnon transports along the <100> direction), an average $\nabla T^W / \nabla T^S$ of 3 times is obtained. Since the heating power is fixed in our experiment, the <100> thermal conductivity for CrPS₄ is estimated to be 3 times smaller than the <010> thermal conductivity, which is 2.1[W/(m*K)].

For the parameters related to spin chemical potential distribution analysis, the ratio for the spin Seebeck coefficient components $S_y:S_x:S_z = 1:0.404:0.017$ are obtained from our model (details in our reply to reviewer's comment #2 and in Fig. R3 below). Here we use the spin Seebeck coefficient 500[A/m] for YIG as S_y for CrPS₄, then applying the ratio above, one can obtain $S_x = 202[\text{A/m}]$ and $S_z = 8.5[\text{A/m}]$. For the magnon spin conductivity σ_m , since the calculation procedure of σ_m is similar to the calculation for \mathbf{S} in our model used in the reply to reviewer's comment #2, we would simply provide the resulting ratio in the following discussion. By adding the chemical potential μ_m generated by SSE caused magnons accumulation under the injector $n_i^0(k) = \frac{1}{e^{\frac{\hbar\omega_i - e\mu_m}{k_B T}} - 1}$ in our model, the spin current density reads:

$$\mathbf{J}_m = \mathbf{S} \cdot \nabla T + \frac{\hbar}{2e} \boldsymbol{\sigma}_m \cdot \nabla \mu_m$$

The resulted magnon spin conductivity is:

$$\boldsymbol{\sigma}_m(T) = \frac{2e^2}{(2\pi)^3 k_B T} \sin\psi \int_{BZ} dk_x dk_y dk_z \sum_{i=1}^2 \mathbf{v}_i(\mathbf{k}) \cosh\delta_i \mathbf{v}_i(\mathbf{k}) \frac{e^{\frac{\hbar\omega_i}{k_B T}} \tau_i(\mathbf{k})}{\left(e^{\frac{\hbar\omega_i}{k_B T}} - 1\right)^2}$$

Thus we could get the ratio for the magnon spin conductivity components $\sigma_{my}:\sigma_{mx}:\sigma_{mz} = 1:0.419:0.037$. Here we use the magnon spin conductivity 9000[S/m] for YIG as σ_{my} for CrPS₄, then $\sigma_{mx} = 3771$ [S/m] and $\sigma_{mz} = 333$ [S/m]. As for the magnon spin diffusion length λ_m , we extracted $\lambda_{my} = 0.87\mu m$ and $\lambda_{mx} = 0.45\mu m$ from $R_{2\omega} = \frac{c_0}{\lambda_m} * \frac{\exp(d/\lambda_m)}{1-\exp(2d/\lambda_m)}$ (Nat. Phys. 11, 1022-1026 (2015)) using $R_{2\omega}^S(d = 0.75\mu m) = 0.527nV/\mu A^2$, $R_{2\omega}^S(d = 1.5\mu m) = 0.189nV/\mu A^2$ for Device-S and $R_{2\omega}^W(d = 0.75\mu m) = 0.352nV/\mu A^2$, $R_{2\omega}^W(d = 1.5\mu m) = 0.064nV/\mu A^2$ for Device-W. The $R_{2\omega}$ values are obtained from the linear slopes of $V_{2\omega,0}$ vs. I_{in}^2 curves for device2 shown in Fig. 2d in our main text and Fig. S3.2d in the Supplementary information. All the parameters used in the finite element analysis are listed below:

Pt	Conductivity	8.9E6[S/m]	COMSOL Material database
	thermal conductivity	71.6[W/(m*K)]	COMSOL Material database
CrPS ₄	in-plane <010> thermal conductivity	6.3[W/(m*K)]	ACS Nano,14, 2424–2435(2020) for MnPS ₃
	in-plane <100> thermal conductivity	2.1[W/(m*K)]	calculated
	through-plane thermal conductivity	1.1[W/(m*K)]	ACS Nano,14, 2424–2435(2020) for MnPS ₃
	in-plane <010> spin Seebeck coefficient	500[A/m]	Phys. Rev. B 96, 104441 (2017) for YIG
	in-plane <100> spin Seebeck coefficient	202[A/m]	calculated
	through-plane spin Seebeck coefficient	8.5[A/m]	calculated
	in-plane <010> magnon spin conductivity	9000[S/m]	Phys. Rev. B 94, 180402(R) (2016) for YIG
	in-plane <100> magnon spin conductivity	3771[S/m]	calculated
	through-plane magnon spin conductivity	333[S/m]	calculated
	in-plane <010> magnon	0.87 μm	From experimental

	spin diffusion length		data
	in-plane $\langle 100 \rangle$ magnon spin diffusion length	$0.45\mu\text{m}$	From experimental data
SiO ₂	thermal conductivity	1.38[W/(m*K)]	CRC Handbook of Chemistry and Physics (92nd ed.).p12.213
Si	thermal conductivity	130[W/(m*K)]	COMSOL Material database
CrPS ₄ /SiO ₂	through-plane thermal resistance	$5\text{E-}7[\text{K}\cdot\text{m}^2/\text{W}]^\#$	Computational Materials Science, 142, 1–6 (2018)
Pt/CrPS ₄	through-plane thermal resistance	$1.4\text{E-}7[\text{K}\cdot\text{m}^2/\text{W}]^\S$	PHYSICAL REVIEW B 101, 205407 (2020)

Table R1. The parameters used in the finite element analysis. [#]There is no data found for CrPS₄/SiO₂ in the literature, we used value from through-plane thermal resistance between MoS₂/SiO₂ instead. [§]There is no data found for Pt/CrPS₄, we used estimated value for CrBr₃/Pt in the literature.

In the simulations, the sample thickness is $t_{\text{CrPS}_4} = 30$ nm and the width of the crystal is $w_{\text{CrPS}_4} = 10$ μm . The crystal is placed on top of a silicon substrate with a SiO₂ layer of 300 nm thick. The injector electrode has a thickness of $t_{\text{Pt}} = 10$ nm and a width of $w_{\text{Pt}} = 250$ nm. The heat current normal to the CrPS₄|vacuum and Pt|vacuum interfaces is set to be zero. The spin current normal to the CrPS₄|vacuum and CrPS₄|SiO₂ interfaces is also set to be zero. We have performed finite element analysis for a MnPS₃ device having similar configuration in *Nat. Commun.* **12**, 6279 (2021), which gives a temperature around 3K at the bottom of the Pt injector. Since here we have adopted the thermal conductivity for MnPS₃ as for CrPS₄, to simplify the simulation, the boundary condition $T = 3\text{K}$ at the bottom of the Pt injector is used.

As shown in Figure R1, under the detector electrodes ($1.5\mu\text{m}$ away from the injectors), both the in-plane temperature gradients and the out-of-plane temperature gradients are negligible for Device-S and Device-W. The large local out-of-plane

temperature gradient should cause a large local SSE signal, and it is indeed detected experimentally, as shown in Fig. S6 in the Supplementary information. However, different from the local signal, the nonlocal signal is mainly caused by the thermal magnons diffusing to the detector. The resultant spin chemical potential distribution has similar behavior to YIG (Phys. Rev. B 96, 104441 2017) that the chemical potential changes sign because of the depletion of magnons below the injector and an accumulation of magnons at the CrPS₄|SiO₂ interface. The finite element analysis shows that out-of-plane temperature gradient plays a minimal role in our nonlocal experiment.

We have put the above finite element analysis in the revised Supplementary Information S11.

Fig. R1 (Fig. S11 in the revised Supplementary Information). Finite element analysis of the temperature and spin chemical potential distribution in CrPS₄ device. (a) Temperature, (b) temperature gradient and (c) spin chemical potential distribution in yz plane for Device-S. (d) Temperature, (e) temperature gradient and (f) spin chemical potential distribution in xz plane for Device-W.

2. Given the above considerations, Equation 2 should be reconsidered carefully.

Equation 2: $V_{ISHE}(I_{in}(t), I_{gate}) \propto g_{mix} \hat{\mathbf{n}} \cdot \mathbf{S}(T) \cdot \nabla T$ describes the inverse spin Hall voltage V_{ISHE} generated by thermal magnons at the detector electrode, which is deduced from a two-dimensional spin transport model only considering the in-plane spin Seebeck coefficients. Based on our finite element analysis, the out-of-plane temperature gradient $\nabla_z T$ is relatively small under the detector. According to the suggestions of the reviewer, we extend our model in Supplementary S8 and S9 to three-dimension and calculate the out-of-plane spin Seebeck coefficient $\mathbf{S}_z(T)$.

Magnetism for the bulk CrPS₄ is described by the following Hamiltonian:

$$H = \sum_{\mathbf{j}} \sum_{m=1,2,3,4} J_{a_m} \mathbf{S}_{\mathbf{j}}^A \cdot \mathbf{S}_{\mathbf{j}+\mathbf{a}_m}^B - D \sum_{\mathbf{j}} [(S_{\mathbf{j}}^{A,z})^2 + (S_{\mathbf{j}}^{B,z})^2] - h \sum_{\mathbf{j}} [S_{\mathbf{j}}^{A,y} + S_{\mathbf{j}}^{B,y}] + \sum_{\mathbf{j}} J_c (\mathbf{S}_{\mathbf{j}}^A \cdot \mathbf{S}_{\mathbf{j}+\mathbf{a}_5}^A + \mathbf{S}_{\mathbf{j}}^B \cdot \mathbf{S}_{\mathbf{j}+\mathbf{a}_5}^B)$$

Here D is the easy-axis single-ion anisotropy and J is the magnetic exchange coupling with $J_{a_1} = J_1$, $J_{a_2} = J_2$, $J_{a_3} = J_3$, $J_{a_4} = J_3$. \mathbf{j} denotes a monoclinic-lattice A-sublattice site, and \mathbf{a}_m ($m = 1,2,3,4,5$) connects a A-sublattice site and its neighboring four B-sublattice sites with $\mathbf{a}_1 = 0$, $\mathbf{a}_2 = \mathbf{e}_2$, $\mathbf{a}_3 = \mathbf{e}_1$, $\mathbf{a}_4 = \mathbf{e}_2 - \mathbf{e}_1$. $\mathbf{a}_5 = \mathbf{e}_3$ is the normal vector connected two adjacent layers. $\mathbf{S}_{\mathbf{j}}^A \equiv (S_{\mathbf{j}}^{A,x}, S_{\mathbf{j}}^{A,y}, S_{\mathbf{j}}^{A,z})$ is a localized spin of Cr atom ($S=3/2$) in the A-sublattice site (\mathbf{j}) and $\mathbf{S}_{\mathbf{j}+\mathbf{a}_m}^B$ is a localized Cr spin at the B-sublattice site($\mathbf{j} + \mathbf{a}_m$).

Under the in-plane field h , the antiferromagnetically aligned spins will be canted linearly towards the field direction:

$$\mathbf{S}_{\mathbf{j}}^A = \mathbf{S}_{\mathbf{j}}^B = S(0, \sin\psi, \cos\psi)|_{j_z=2n\mathbf{e}_3}$$

$$\mathbf{S}_{\mathbf{j}}^A = \mathbf{S}_{\mathbf{j}}^B = S(0, \sin\psi, -\cos\psi)|_{j_z=(2n+1)\mathbf{e}_3}$$

with $2n$ denotes even layers and $2n+1$ denotes odd layers. A canting angle is

determined classically as a minimum of a classical magnetic energy:

$$E_{classical} = N(2J_c S^2 (\sin^2 \psi - \cos^2 \psi) - 2DS^2 \cos^2 \psi - 2hS \sin \psi),$$

where N is a number of the A-sublattice sites. The minimum energy is given by:

$$\sin \psi = \frac{h}{2(2J_c + D)S}$$

Magnetic collective excitations around the classical magnetic order are described by the Holstein-Primakoff bosons:

$$\begin{aligned} \tilde{S}_j^{A,z} &= S - a_j^\dagger a_j, & \tilde{S}_j^{A,x} - i\tilde{S}_j^{A,y} &= \sqrt{2S}a_j^\dagger, & \tilde{S}_j^{A,x} + i\tilde{S}_j^{A,y} &= \sqrt{2S}a_j \\ \tilde{S}_j^{B,z} &= S - b_j^\dagger b_j, & \tilde{S}_j^{B,x} - i\tilde{S}_j^{B,y} &= \sqrt{2S}b_j^\dagger, & \tilde{S}_j^{B,x} + i\tilde{S}_j^{B,y} &= \sqrt{2S}b_j \end{aligned}$$

where $(\tilde{S}_j^{A,x}, \tilde{S}_j^{A,y}, \tilde{S}_j^{A,z})$ and $(\tilde{S}_j^{B,x}, \tilde{S}_j^{B,y}, \tilde{S}_j^{B,z})$ are the spin operators in a rotated frame:

$$\begin{aligned} \begin{pmatrix} \tilde{S}_j^{\alpha,x} \\ \tilde{S}_j^{\alpha,y} \\ \tilde{S}_j^{\alpha,z} \end{pmatrix} &= \begin{pmatrix} 1 & 0 & 0 \\ 0 & \cos\psi & -\sin\psi \\ 0 & \sin\psi & \cos\psi \end{pmatrix} \begin{pmatrix} S_j^{\alpha,x} \\ S_j^{\alpha,y} \\ S_j^{\alpha,z} \end{pmatrix} \Big|_{j_z=2n\mathbf{e}_3} \\ \begin{pmatrix} \tilde{S}_j^{\alpha,x} \\ \tilde{S}_j^{\alpha,y} \\ \tilde{S}_j^{\alpha,z} \end{pmatrix} &= \begin{pmatrix} 1 & 0 & 0 \\ 0 & -\cos\psi & -\sin\psi \\ 0 & \sin\psi & -\cos\psi \end{pmatrix} \begin{pmatrix} S_j^{\alpha,x} \\ S_j^{\alpha,y} \\ S_j^{\alpha,z} \end{pmatrix} \Big|_{j_z=(2n+1)\mathbf{e}_3} \end{aligned}$$

Here $\alpha = A, B$, a and b are Holstein-Primakoff boson fields for A-sublattice Cr spins and B-sublattice Cr spins, that represent fluctuations around the classical magnetic order. Around those ψ that minimize the classical magnetic energy, the Hamiltonian is stable against such small fluctuations:

$$H \equiv E_{classical} + H_{sw} + \mathcal{O}(a^3, b^3)$$

Thus, the spin-wave Hamiltonian transformed into the momentum space (here \mathbf{k} is a three-dimensional wave vector) is:

$$H_{magnon} = \sum_{\mathbf{k}} \Psi^\dagger(\mathbf{k}) \begin{bmatrix} M_0 & f(\mathbf{k}) & N_0 & 0 \\ f^*(\mathbf{k}) & M_0 & 0 & N_0 \\ N_0 & 0 & M_0 & f(\mathbf{k}) \\ 0 & N_0 & f^*(\mathbf{k}) & M_0 \end{bmatrix} \Psi(\mathbf{k})$$

with $\Psi(\mathbf{k}) = (a(\mathbf{k}), b(\mathbf{k}), a^\dagger(-\mathbf{k}), b^\dagger(-\mathbf{k}))^T$, $M_0 = \frac{-J_1 S - J_2 S - 2J_3 S}{2} + DS \cos^2 \psi - \frac{DS}{2} \sin^2 \psi + \frac{h}{2} \sin \psi + J_c S \cos 2\psi + \frac{J_c S}{2} (1 - \cos 2\psi) \cos(\mathbf{k} \cdot \mathbf{a}_5)$, $N_0 = \frac{DS}{2} \sin^2 \psi +$

$\frac{JcS}{2}(1 + \cos 2\psi)\cos(\mathbf{k} \cdot \mathbf{a}_5)$, $|f(\mathbf{k})|$ and φ_k are the modulus and phase of $f(\mathbf{k}) = \frac{S}{2}\sum_{m=1,2,3,4}J_{a_m}e^{ik \cdot \mathbf{a}_m} \equiv |f(\mathbf{k})|e^{i\varphi_k}$ respectively.

Under

$$\begin{pmatrix} a(\mathbf{k}) \\ b(\mathbf{k}) \\ a^+(-\mathbf{k}) \\ b^+(-\mathbf{k}) \end{pmatrix} = \sigma_0 \otimes \frac{1}{\sqrt{2}} \begin{pmatrix} e^{i\frac{\varphi_k}{2}} & 0 \\ 0 & e^{-i\frac{\varphi_k}{2}} \end{pmatrix} \begin{pmatrix} 1 & -1 \\ 1 & 1 \end{pmatrix} \begin{pmatrix} \alpha_k \\ \beta_k \\ \alpha_{-k}^\dagger \\ \beta_{-k}^\dagger \end{pmatrix}$$

$$\begin{pmatrix} \alpha_k \\ \alpha_{-k}^\dagger \end{pmatrix} = \begin{pmatrix} \cosh \frac{\delta_1}{2} & -\sinh \frac{\delta_1}{2} \\ -\sinh \frac{\delta_1}{2} & \cosh \frac{\delta_1}{2} \end{pmatrix} \begin{pmatrix} \gamma_{1,k} \\ \gamma_{1,-k}^\dagger \end{pmatrix}, \begin{pmatrix} \beta_k \\ \beta_{-k}^\dagger \end{pmatrix}$$

$$= \begin{pmatrix} \cosh \frac{\delta_2}{2} & \sinh \frac{\delta_2}{2} \\ \sinh \frac{\delta_2}{2} & \cosh \frac{\delta_2}{2} \end{pmatrix} \begin{pmatrix} \gamma_{2,k} \\ \gamma_{2,-k}^\dagger \end{pmatrix}$$

The spin-wave Hamiltonian is diagonalized as,

$$H_{magnon} = \sum_{\mathbf{k}} (\hbar\omega_1(\mathbf{k})\gamma_1^\dagger(\mathbf{k})\gamma_1(\mathbf{k}) + \hbar\omega_2(\mathbf{k})\gamma_2^\dagger(\mathbf{k})\gamma_2(\mathbf{k}))$$

Here,

$$\cosh \delta_1 = \frac{M_0 + |f(\mathbf{k})|}{E_1}$$

$$\cosh \delta_2 = \frac{M_0 - |f(\mathbf{k})|}{E_1}$$

$$E_1(\mathbf{k}) = \hbar\omega_1(\mathbf{k}) = \sqrt{(M_0 + |f(\mathbf{k})|)^2 - N_0^2}$$

$$E_2(\mathbf{k}) = \hbar\omega_2(\mathbf{k}) = \sqrt{(M_0 - |f(\mathbf{k})|)^2 - N_0^2}.$$

The lowest E_2 spin-wave energy momentum dispersions with $k_y = 0$ or $k_x = 0$ are plotted in Fig. R2. As can be seen, the energy band is nearly dispersionless along k_z direction compared to k_x and k_y .

Fig. R2(Fig. S10.1 in the revised Supplementary Information). Three-dimensional spin model and spin wave modes of CrPS₄ under an in-plane magnetic field. (a) The lowest E_2 spin-wave energy momentum dispersions with $k_y = 0$. (b) The lowest E_2 spin-wave energy momentum dispersions with $k_x = 0$.

To get the expression for spin Seebeck coefficient, we first consider the experimental configuration for Device-S, where the magnetic field is along the y axis. The spin density along the field direction is given by magnon creation and annihilation operators:

$$\sum_j (S_j^{A,y} + S_j^{B,y}) = \sin\psi \sum_j (2S - a_j^\dagger a_j - b_j^\dagger b_j) + \cos\psi \sum_j \sqrt{\frac{S}{2}} (a_j^\dagger + a_j + b_j^\dagger + b_j)$$

We consider the contribution of the first term (the second linear term vanishes in average) and obtain the average spin projection along the magnetic field as:

$$\sum_j \langle S_j^{A,y} + S_j^{B,y} \rangle = -\sin\psi \sum_{\mathbf{k}} \langle \cosh\delta_1 \gamma_1^\dagger(\mathbf{k}) \gamma_1(\mathbf{k}) + \cosh\delta_2 \gamma_2^\dagger(\mathbf{k}) \gamma_2(\mathbf{k}) \rangle$$

With $n_i(\mathbf{k}) = \langle \gamma_i^\dagger(\mathbf{k}) \gamma_i(\mathbf{k}) \rangle$, the spin current density operator J_m is:

$$J_S^y = -\frac{\hbar}{(2\pi)^3} \sin\psi \int dk_x dk_y dk_z \sum_{i=1}^2 v_i(\mathbf{k}) \cosh\delta_i [n_i(\mathbf{k}) - n_i^0(\mathbf{k})]$$

$$n_i(\mathbf{k}) - n_i^0(\mathbf{k}) = -\tau_i(\mathbf{k}) v_i(\mathbf{k}) \cdot \nabla n_i^0(\mathbf{k})$$

For Device-W, a similar J_S^x could be obtained. Based on $J_m = \mathbf{S} \cdot \nabla T$, the three-dimensional spin Seebeck coefficient reads:

$$\mathbf{S}(T) = \frac{\hbar}{(2\pi)^3 k_B T^2} \sin\psi \int_{BZ} dk_x dk_y dk_z \sum_{i=1}^2 \mathbf{v}_i(\mathbf{k}) \cosh\delta_i \mathbf{v}_i(\mathbf{k}) \frac{e^{\frac{\hbar\omega_i}{k_B T}} \hbar\omega_i \tau_i(\mathbf{k})}{\left(e^{\frac{\hbar\omega_i}{k_B T}} - 1\right)^2}$$

Figure R3 shows the calculated spin Seebeck coefficient $\mathbf{S}_n(T)$. The out-of-plane spin Seebeck coefficient $\mathbf{S}_z(T)$ is quite small compared with the in-plane coefficients $\mathbf{S}_x(T)$ and $\mathbf{S}_y(T)$ at $T = 2\text{K}$ and above. This 2D nature of the magnon transport in CrPS₄ is warranted by the weak interlayer exchange interaction.

Fig. R3 (Fig. S10.2 in the revised Supplementary Information). The simulated spin Seebeck coefficient \mathbf{S}_n with a three-dimensional model.

Since both of the out-of-plane temperature gradient under the detector and the spin Seebeck coefficient $\mathbf{S}_z(T)$ are small, the two-dimensional spin transport model could capture the majority of the physics and thus the Equation 2 is a good approximation at the moment. It is indeed interesting to provide a more comprehensive theory to the experimental result we obtained in this manuscript, and we hope that our experimental work could stimulate further discussions and theoretical works in the field.

According to the reviewer's comments, we have put the above discussions in the revised Supplementary Information S10. And we have added discussion in page 10 line 217 of the revised main text: "Finite element analysis³⁶ and three-dimensional spin model were performed, and the effect of thermal gradient and spin Seebeck

coefficient along the z direction is estimated to be small (details in Supplementary Information S10-S11).”

3. In the abstract: 2500000% is oversold and misleading. It sounds as if it were an on-off ratio. It is not fair unless it is clearly written that the data is measured by different devices on the same film.

We thank the reviewer for bring up this point that might cause misunderstanding. Indeed the anisotropic ratio is not an on-off ratio of conventional meaning, e.g. that the same device be turned on and off with different electric gate. Rather, it tells us how precisely we can control the flow of information (magnon signal) in on direction but stop the transmission of such information in the other direction.

Additional, we share the same view with the reviewer that the value of the anisotropic ratio is meaningful only when we compare two magnon devices (one along the strong axis and the other along the weak axis) fabricated on the same single crystal, through the same microfabrication process, and measured in the same circuit under the same experiment. For the more than 10 pairs of devices, this is exactly what we have done.

We would like to stress that when the signal of Device-W goes from positive to negative, the zero point is guaranteed, thus the anisotropy ratio $|V_{2\omega,0}^S/V_{2\omega,0}^W|$ should be a point of divergence. However, the experimental value we could get is influenced by the step size of I_{gate} sweep and the noise floor of measurement as we mentioned in page 9, line 206-213 of the manuscript. Thus, the anisotropy ratio of 2,500,000% is the noise-limited measured value we could get experimentally. To avoid the possible confusion, we have modified our abstract in page 2, line 37 to be: “Here, we realized giant electrically tunable anisotropy in the transport of second harmonic thermal magnons (SHM) in van der Waals anti-ferromagnetic insulator CrPS₄ with the application of modest gate current.”

(minor points)

4. Line 190: A brief summary of the contents of the SI S6-7 may be added to the main text for better readability.

We appreciate the helpful suggestion of the reviewer. We have added the following description in page 9, line 193: “In particular, the local spin Seebeck signal of the injector electrode could not be tuned to inverse as a function of I_{gate} , which shows that the magnon diffusive process and anisotropic magnetic exchange interactions are vital in producing the highly tunable anisotropic nonlocal signal. Furthermore, we measured the non-local second harmonic signal with an applied magnetic field of up to 9T rotated in the x - z plane. The signal is almost zero when the magnetic field is along the z axis, which indicates the absence of an anomalous Nernst effect (details in Supplementary Information S6-S7).” The added discussion is marked in blue in the revised manuscript.

5. Line 262: Please add an explanation on the expansion of Equation 4.

To make the expansion of Equation 4 more clear, we have modified our manuscript in page 12, line 268 to be: “The temporal dependence of V_{ISHE} comes purely from the time variation of $I_{in}^2(t) \propto \sin^2(\omega t)$. By substituting T in $\mathbf{S}(T)$ with $2K + c_1 I_{in}^2 + c_2 I_{gate}^2$, we can use the following equation to fit our $V_{2\omega,0}^S$ and $V_{2\omega,0}^W$ data (labeled as $V_{2\omega,0}^{S,W}$):

$$\begin{aligned} V_{2\omega,0}^{S,W} &= C^{S,W} * [\hat{\mathbf{n}} \cdot \mathbf{S}(T) \cdot \nabla T]_{2\omega} \\ &= C^{S,W} * [S_n(2K + c_1^{S,W} I_{in}^2 + c_2^{S,W} I_{gate}^2) * (c_1^{S,W} I_{in}^2 + c_2^{S,W} I_{gate}^2)]_{2\omega} \end{aligned} \quad (4)$$

where $V_{2\omega,0}^S$ ($V_{2\omega,0}^W$) is the SHM signal of Device-S (Device-W), $C^{S,W}$ is a global parameter containing g_{mix} , S_n is the spin Seebeck coefficient component along $\hat{\mathbf{n}}$ direction, $c_1^{S,W}$ and $c_2^{S,W}$ are heating efficiencies of the injector and gate along the

two directions, respectively, and $[...]_{2\omega} = \frac{\omega}{\pi} \int_{-\frac{\pi}{\omega}}^{\frac{\pi}{\omega}} \cos(2\omega t) * [...] dt$ means taking the second harmonic component.”

6. Page 14- The ROM part does not provide new physics but the ROM has not been analyzed for its engineering merits from application points of view: the ROM seems too slow and too large (it is difficult to confine heat current into small area). This part can be compressed.

We agree with the reviewer that the magnon ROM we displayed is a prototypical application based on the electrically tunable anisotropic magnon transport, rather than a commercial device that outperformed electron-based devices. The importance of such magnon ROMs lies in the fact that it points to device concept and offer new possibilities for future special purposed information storage such as inscribing proprietary information and confidential information. We believe that future engineering along the line of this new concept would likely lead to improved performance including faster operation speed, lower power consumption, smaller device sizes, etc.

To reflect the reviewer’s comment, we have deleted the description in page 16, line 349 in the main text: “It is worth noting that with its peculiar readout scheme, such magnon ROMs can serve in special purposed information storage such as inscribing proprietary and confidential information. That is to say, the information stored in such magnon ROMs is out of reach for persons without a prior knowledge of several factors including the Néel temperature of the channel materials, the channel crystal orientation and magnetization direction, the preset I_{gate} , etc. The anisotropic magnon ROM could also be used to store two sets of digital information, which can be read out using two different gate currents (i.e., $I_{gate} = I_0^S$ and $I_{gate} = I_0^W$ represent two sets of information for the same ROM). Multi-state (instead of binary) memory can in principle be engineered by making use of the nonlinear and anisotropic relation

between I_{gate} and I_{read} along the two directions.” We have moved such discussion to supplementary information S15 for further reading of interested readers.

Reviewer #2 (Remarks to the Author):

In this manuscript, the authors report anisotropy in diffusive magnon transport in van Der Waals (vdW) antiferromagnetic insulator CrPS4. This was accomplished by performing nonlocal magnon transport measurement along different crystal axes. The authors demonstrate that anisotropy is tunable via a gate current, with an anisotropy ratio of up to 2500x achieved. Lastly, by utilizing the observed anisotropic magnon transport, the authors demonstrate a read-only memory (ROM) whose value is set by placement of read electrodes.

The experiment is well done; the results and their interpretations are convincing. The scientific claims are well supported, and methodology is sound. The work should be reproducible given the details provided in the manuscript.

We are grateful that the reviewer pointed out the major findings of our work and have positively commented on the scientific rigor of our results.

However, I am not convinced that the impact of the result is sufficient to warrant publication in Nature Communications. In the introduction, the authors have cited recent interest in various types of anisotropy, which supposedly is to argue for the significance of magnon-transport anisotropy studied in this work. Indeed, electrical and optical anisotropy have various anticipated applications as discussed in the literature, such as directional memory (Ref. 1), polarization-sensitive photodetectors (Refs. 4, 5), novel polarizers/polarization sensors (Xia, Wang, Jia, Nat. Commun. 5, 4458 (2014)), to name a few. However, it is not clear that there is high-impact device applications envisioned that relies on anisotropic diffusive magnon-transport studied in this work. There is a statement "In particular, in-situ electrical modulation of

anisotropy in spin transport, a vital functionality for future large-scale applications of van der Waals magnets, has yet to be achieved"; why electrical modulation of anisotropy is vital to large-scale applications of vdW magnets is neither explained nor supported by citation. ROM based on anisotropic magnon transport is interesting, but it does not have prospect for future practical applications. In particular, this ROM requires a large field (4 T) which is not practical. Lastly, the electrical tunability relies on heating coming from gate current, which is likely not fast and possibly highly temperature-dependent, hence less of an ideal control modality compared to electric field gating (if it could be used to control anisotropy).

We appreciate the reviewer's comment which points to the need for us to better explain the importance of our findings. Our reply to the above comments shall be listed in the five points below:

1) The importance of anisotropy in materials in general:

We share the same view with the reviewer that anisotropy has various anticipated applications, which motivated us to do this research. We also thank the reviewer for providing a good supporting literature (Nat. Commun. 5, 4458 (2014)) about applications of optical anisotropy. We have added this reference in page 3, line 55 as: "Anisotropic light-matter interactions in 2D materials also facilitate applications in various optoelectronic devices like novel polarizers or polarization sensors⁴, polarized light-emitting diodes and polarization-sensitive photodetectors^{5, 6}."

2) The importance of magnon transport anisotropy:

First, magnetic anisotropy participate in the formation of stable magnetic structures and thus plays a pivotal role in enabling spin-based information technology, leading to important applications such as tunneling spin valves. Magnon transport anisotropy is an important and more recent development of magnetic anisotropy, where the focus is on the anisotropy of the transport of spin fluctuations instead of the orientation of steady spin structure.

3) The importance of electrical tunability:

Magnetic field is the most natural knob for controlling the magnetic structure and spin excitations, including magnons; however, magnetic field is very difficult to localize, making magnetically controlled large-scales integrated spintronics circuits difficult to realize. Thus, electrical control of magnetic properties becomes an important direction both scientifically and technologically.

Electrical modulation of anisotropy including the easy magnetization direction and anisotropic magnetoresistance is vital for spintronics study (J. Magn. Magn. Mater. 563, 169924 (2022), Annu. Rev. Mater. Res. 44, 91-116 (2014)). For the applications of van der Waals magnets as well as for applications in spin transport, the same logic applies. Here we realize the electrically tunable anisotropy of diffusive magnon transport, which could enable vital applications like storage and logic operation based on 2D magnons in van der Waals magnets. To reflect the comment of the reviewer, we have modified “In particular, *in-situ* electrical modulation of anisotropy in spin transport, a vital functionality for future large-scale applications of van der Waals magnets, has yet to be achieved.” in page 3, line 62 to be: “As electrically controlled magnetic anisotropy plays an important role in spintronics^{17, 18}, *in-situ* electrical modulation of anisotropy in spin transport could also enable vital applications like storage and logic operation based on 2D magnons in van der Waals magnets. However, such *in-situ* electrical modulation has yet to be achieved.”

4) The magnon transport anisotropy based ROM:

Here the ROM we displayed is a prototypical application based on the tunable anisotropic magnon transport, which is important as a new concept device that is based on completely new physical principles. We are confident that future engineering along the line of this new concept we pointed out will surely lead to improved performance including zero field operation, lower power consumption, smaller device sizes, etc. Besides information storage, *in-situ* electrical modulation of anisotropy in

spin transport could also enable vital functions of logic operation and information transmission like duplexers and even routers based on diffusive magnons.

5) Important advancement in spin caloritronics:

We also agree with the reviewer's point: "Lastly, the electrical tunability relies on heating coming from gate current, which is likely not fast and possibly highly temperature-dependent, hence less of an ideal control modality compared to electric field gating (if it could be used to control anisotropy)." Electric field gating tunable magnon transport is more attractive than current, which is also a challenge so far. We are actively working along this direction. Nevertheless, spin caloritronics is also an invigorated field both for scientific curiosity and for potential applications (Nat. Mater. 11, 391-399 (2012)). Our work reported the anisotropic nonlocal spin Seebeck effect for the first time and developed a prototypical application, which is an important achievement in spin caloritronics that has never been achieved in either van der Waals magnets or conventional magnets.

Given the above consideration, I cannot recommend publication in a high impact journal for diverse audience such as Nature Communications. I would recommend publication in journals more specific to materials or nano science; for example, this manuscript would be suitable for NPJ Quantum Materials or NPJ 2D Materials and Applications.

We respectfully argue that our work is highly important because we realized electrically controlled anisotropic magnon transport for the first time, and the resulted anisotropic ratio is much larger than the reported ones like magnetic field tuned magnon transport anisotropy of ~200% reported in α -Fe₂O₃ thin film (Nat. Nanotechnol. 15, 563-568 (2020)) and crystalline anisotropy induced magnon transport anisotropy of ~150% reported in MgAl_{0.5}Fe_{1.5}O₄ thin film (Nano Lett. 22, 1167-1173 (2022)). What's more, our work is a discovery of a general and highly tunable anisotropic response which could be applied to a large class of van der Waals

magnets with in-plane anisotropy. We believe our work represents a breakthrough in the field of low symmetry spintronics and van der Waals electronic devices, which will be of great interest to a diverse audience including scientists working in the physical sciences, spintronics, nanotechnology, and material science.

In addition, here I also list a few other comments:

We appreciate the reviewer's comments that help us substantially improve our manuscript.

- when discussing magnons in 2D magnets (around line 61), it would be appropriate to also cite <https://www.nature.com/articles/s41586-022-05024-1>

We would like to thank the reviewer for reminding us of this reference. Indeed this paper is an important advance in the study of van der Waals magnons and is highly relevant to our work. In particular, this paper provided an optical pump-probe technique to excite coherent magnons via their interactions with optically created excitons; while our results report electrically excited diffusive magnons whose transport anisotropy can be extensively and reversibly tuned. According to the reviewer's suggestion, we have added this reference in page 3, line 61 as: "So far, highly tunable 2D magnons has been demonstrated in graphene quantum ferromagnet/anti-ferromagnet^{11, 12}, layered anti-ferromagnet CrI₃^[13, 14], CrSBr^[15] and MnPS₃^[16], but in-plane anisotropic properties of 2D magnets are much less explored."

-there is this statement in line 87-90: "Furthermore, ferromagnetism with out-of-plane spin orientation is found in monolayer, air-stable CrPS4 crystals, making CrPS4 more attractive candidate for applications in van der Waals spintronics"

This sentence is confusing. What is "more attractive" compared to? Perhaps the author

simply meant to say CrPS₄ is "attractive", or "rather attractive". Also, it's not clear what are the features that the authors consider to make this material attractive for spintronics.

We thank the reviewer for pointing out this sentence where the language needs to be improved. In addition to in-plane anisotropy, the most important feature making this material attractive for spintronics is the A-type antiferromagnetic structure as we mentioned in page 4, line 85 in our manuscript. Such magnetic structure intrinsically acts as a magnetic tunnel barrier like CrI₃ (Science 360, 1214-1218 (2018), Science 360, 1218-1222 (2018)), which is the building block of spintronics devices like data storage and magnetic sensors. In addition, since some of the 2D magnets are very sensitive to air when they are exfoliated to thin films, e.g. CrI₃, CrCl₃, VI₃ and Cr₂Ge₂Te₆, the air-stable CrPS₄ is “more attractive” for applications compared to these materials. What’s more, monolayer CrPS₄ could remain magnetically ordered (ferromagnetic) and is also air-stable, which is a promising feature for developing spintronics devices down to the atomic limit. Nevertheless, it’s not necessary to emphasize the type of magnetic orders in monolayer CrPS₄, and the language of the sentence needs improvement. Thus we have adopted the suggestion of the reviewer and modified page 4, line 93 in the revised manuscript to be: “Furthermore, CrPS₄ crystals are air stable and magnetically ordered down to the monolayer limit²³, making CrPS₄ rather attractive candidate for applications in van der Waals spintronics.”

-Why not replace Fig. 3c with the inset, which shows giant tunability of anisotropy?

We thank the reviewer for the thoughtful suggestion. The current version of the main panel of Fig. 3c shares the same experimental and simulation data taken from one device as in Fig. 3a. If we switch the main panel of Fig. 3c into a small inset, it would be difficult to clearly see the degree of agreement between the theoretical simulation and the experiment data. What’s more, the current version of Fig. 3c illustrates the

existence of the divergence points in the anisotropy ratio $|V_{2\omega,0}^S/V_{2\omega,0}^W|$, which is of important physical and practical meaning. On the other hand, the current version of inset in Fig. 3c is a case of the noise-limited and device quality-dependent measured value we got experimentally from another device. It shows the best we have achieved towards the theoretical prediction of diverging anisotropy ratio, but the specific value of largest anisotropy ratio we measured is basically a reflection of noise floor of our measurement and of the device quality. Thus, we consider the current main panel of Fig. 3c to be the more physical and intrinsic result we would like the readers to clearly see. According to the reviewer's comment, we have changed the labels of Fig. 3c so that the size of the inset can increase and be clearer.

-Line 279: what does "Specifically, 1" refer to?

We thank the reviewer for pointing out this sentence where the language may be difficult for the readers.

As written in page 7, line 159: "As can be seen from Fig. 2a-2d, the overall SHM signal $V_{2\omega}$ is higher for Device-S compared to Device-W, no matter how the parameters T , B or I_{in} varied. This intrinsic anisotropic response is consistently observed in all devices we studied, which manifests the profound effects of structural anisotropy to the magnonic spin transport properties of CrPS₄, and is already useful for applications as-is." We first reports an intrinsic anisotropic SHM signal $V_{2\omega}$ when $I_{gate} = 0$. Then we discovered that such intrinsic anisotropic response originates from the anisotropic spin Seebeck coefficient tensor \mathcal{S} . We understand that the current language is a little difficult to read, and we have rewritten the relative sentences in page 13 line 288 as: "According to the model, the key to the realization of electrically tunable magnon transport anisotropy in CrPS₄ comes from the anisotropic spin Seebeck coefficient tensor \mathcal{S} shown in Eqn.3. There are three specific implications of such \mathcal{S} . First, S_{yy} is larger than S_{xx} under the same excitation with

zero I_{gate} , resulting in stronger signal for Device-S than Device-W before gating (i.e., $I_{gate} = 0$). Here S_{yy} and S_{xx} is the spin Seebeck coefficient matrix element along the $\langle 010 \rangle$ direction and $\langle 100 \rangle$ direction, respectively (see Fig. 2 & 3a, Supplementary Figure S3 & S5). Second, I_0^S is much larger than I_0^W . Here I_0^S and I_0^W are the zero points at the $V_{2\omega,0}^S(I_{gate})$ and $V_{2\omega,0}^W(I_{gate})$ curves, respectively (see Fig. 3a and Supplementary Figure S4, S10a-10d). Third, the negative maximum of $V_{2\omega,0}^W(I_{gate})$ curves are proportionally much larger than that of $V_{2\omega,0}^S(I_{gate})$ curves (see Supplementary Figure S4 and Fig. S10a-10d).”

- it would appropriate to include scale bars in the optical images showing the devices

We thank the reviewer for pointing out the need to include scale bars in the optical images of devices. We have modified Fig. 1 and Fig 4 accordingly.

Reviewers' Comments:

Reviewer #1:

Remarks to the Author:

In the new version, the authors have revised the wording and data presentation appropriately in response to the referees' comments. Now I can recommend this paper for publication.

Reviewer #2:

Remarks to the Author:

The authors have satisfactorily responded comments from my review. I can now recommend publication. I have only two remaining minor points for the authors to consider to help improve the manuscript:

- I realize I don't understand this sentence from line 66-68 on p.3 "Here, we report the realization of electrically tunable anisotropy of diffusive magnon transport in the range of 100% (isotropic) and over 2,500,000% in van der Waals antiferromagnetic insulator CrPS4." I understand the number of 2,500,000%; what does the 100% refer to? Furthermore, it is confusing to see both phrase "in the range of 100%" and "over 2,500,000%" describing tunability in the same sentence. I suggest the authors consider rephrasing this sentence to make it more clear.

- Fig 3: use same axis scaling for main panel and inset, e.g. 10^3 % is now used for the main panel, so perhaps the authors would also like to use the same for the inset.

Reviewer #1 (Remarks to the Author):

In the new version, the authors have revised the wording and data presentation appropriately in response to the referees' comments. Now I can recommend this paper for publication.

We are glad that the reviewer considers our revision satisfactory and recommends the publication of our current manuscript in Nature Communications. We are also grateful for the insightful comments raised by the reviewer, which have prompted us to substantially improve our manuscript.

Reviewer #2 (Remarks to the Author):

The authors have satisfactorily responded comments from my review. I can now recommend publication. I have only two remaining minor points for the authors to consider to help improve the manuscript:

We are glad that the reviewer considers our responses satisfactory and recommends the publication of our current manuscript in Nature Communications. We appreciate the reviewer's comments that help us greatly improve our manuscript. Our point-by-point reply to the additional comments of the reviewer is listed below. All revisions in this round of review process are marked in blue in the revised manuscript.

- I realize I don't understand this sentence from line 66-68 on p.3 "Here, we report the realization of electrically tunable anisotropy of diffusive magnon transport in the range of 100% (isotropic) and over 2,500,000% in van der Waals antiferromagnetic insulator CrPS4." I understand the number of 2,500,000%; what does the 100% refer to? Furthermore, it is confusing to see both phrase "in the range of 100%" and "over 2,500,000%" describing tunability in the same sentence. I suggest the authors consider

rephrasing this sentence to make it more clear.

We thank the reviewer for pointing out this sentence where the language needs to be improved. While “2,500,000%” means a huge anisotropy, “100%” means the isotropy for the two crystallographic directions, i.e., $|V_{2\omega,0}^S| = |V_{2\omega,0}^W|$ with an appropriate I_{gate} . To avoid the possible confusion, we have rephrased our manuscript in page 3, line 66 to be: “Here, we report the realization of electrically tunable anisotropy of diffusive magnon transport from isotropic to an anisotropy ratio of over 2,500,000% in van der Waals antiferromagnetic insulator CrPS₄.”

- Fig 3: use same axis scaling for main panel and inset, e.g. $10^3\%$ is now used for the main panel, so perhaps the authors would also like to use the same for the inset.

We thank the reviewer for the thoughtful comment. Since the data range in the inset of Fig. 3c is much larger than the data range in the main panel, here we use log scale to show the data details clearly instead of linear scale. If we also use $10^3\%$ as unit for the inset, the tick labels will be “0.1, 10, 10^3 ”, which looks a little weird. Thus, we prefer to use the current axis scaling for the inset.